# The cortical connectivity of the periaqueductal gray and the conditioned response to the threat of breathlessness

Olivia K Faull[1,2]*, Kyle TS Pattinson[1,2]

[1]FMRIB Centre, University of Oxford, Oxford, United Kingdom; [2]Nuffield Division of Anaesthetics, Nuffield Department of Clinical Neurosciences, University of Oxford, Oxford, United Kingdom

**Abstract** Previously we observed differential activation in individual columns of the periaqueductal grey (PAG) during breathlessness and its conditioned anticipation (*Faull et al., 2016b*). Here, we have extended this work by determining how the individual columns of the PAG interact with higher cortical centres, both at rest and in the context of breathlessness threat. Activation was observed in ventrolateral PAG (vlPAG) and lateral PAG (lPAG), where activity scaled with breathlessness intensity ratings, revealing a potential interface between sensation and cognition during breathlessness. At rest the lPAG was functionally correlated with cortical sensorimotor areas, conducive to facilitating fight/flight responses, and demonstrated increased synchronicity with the amygdala during breathlessness. The vlPAG showed fronto-limbic correlations at rest, whereas during breathlessness anticipation, *reduced* functional synchronicity was seen to both lPAG and motor structures, conducive to freezing behaviours. These results move us towards understanding how the PAG might be intricately involved in human responses to threat.

**\*For correspondence:** olivia.faull@ndcn.ox.ac.uk

## Introduction

Breathlessness is a complex, multi-dimensional sensation that is a major cause of suffering in people with a diverse range of illnesses, including chronic lung disease (e.g. asthma, chronic obstructive pulmonary disease), cardiac disease and cancer. Subjective perceptions of breathlessness are not simply a neurological reflection of lung function (*Banzett et al., 2000*; *Teeter and Bleecker, 1998*), and can cause crippling anxiety when perceived as a direct threat to survival. These perceptions rely on an intricate system of sensory and affective components within the brain (*Hayen et al., 2013b*; *Herigstad et al., 2011*; *Lansing et al., 2009*), both perceiving ventilatory afferents and evaluating the threat value of these sensations. Furthermore, repeated episodes of breathlessness can lead to conditioned associations between a stimulus (such as a flight of stairs) and breathlessness (*De Peuter et al., 2004*), inducing anticipatory threat behaviours such as freezing or activity avoidance (*Hayen et al., 2013b*).

The midbrain periaqueductal gray (PAG) is a key neural component in both the behavioural modulation of breathing (*Subramanian, 2013*; *Subramanian et al., 2008*) and coordination of threat responses (*Bandler et al., 2000*; *Bandler and Shipley, 1994*; *Keay and Bandler, 2001*; *Tovote et al., 2016*). The PAG is divided into four longitudinal columns (ventrolateral, vlPAG; lateral, lPAG; dorsolateral, dlPAG; dorsomedial, dmPAG) on each side, and is postulated to act as an interface for behavioural control (*Bandler et al., 2000*; *Bandler and Shipley, 1994*; *Benarroch, 2012*; *Keay and Bandler, 2001*). The PAG has previously been implicated in many basic survival functions including cardiovascular, motor and pain responses such as vocalization or blood pressure regulation (*De Oca et al., 1998*; *Mobbs et al., 2007*; *Paterson, 2014*; *Pereira et al., 2010*; *Tracey et al.,*

*2002*). An integrative theory of these behaviours postulates that the lPAG and dlPAG are thought to orchestrate 'active' responses (such as fight or flight responses) when a threat is perceived as escapable (*Bandler and Carrive, 1988*; *Carrive, 1993*; *Depaulis et al., 1992*; *Keay and Bandler, 2001*; *Yardley and Hilton, 1986*). Conversely, the vlPAG is thought to be involved with 'freezing' type behaviours from inescapable threats (*Carrive and Bandler, 1991*; *Keay et al., 1997*; *Lovick, 1993*; *Tovote et al., 2016*), including conditioned anticipation of breathlessness in humans (*Faull et al., 2016b*). In recent work to understand this circuitry in animals, Tovote and colleagues (*Tovote et al., 2016*) intricately mapped direct neuronal connections from the central nucleus of the amygdala to the vlPAG, and onwards to premotor areas in the ponto-medullary reticular formation to underlie freezing behaviours. However, the extent to which this descending circuitry and the additional (unmapped) ascending circuitry between the PAG columns and cortex exist in humans is not yet understood.

Our previous work has functionally subdivided the human PAG with high resolution (7 Tesla) functional magnetic resonance imaging (fMRI). This work revealed that activity can be localised to the lPAG during breath holds (*Faull et al., 2015*), to the vlPAG during conditioned anticipation of breathlessness, and to the lPAG during breathlessness itself (*Faull et al., 2016b*). In this study we further tested the hypothesis that the vlPAG is involved with learned anticipatory threat detection, whilst the lPAG is associated with the active response to breathlessness, by investigating how these PAG regions interact with the wider cortex. We firstly conducted an ultra-high field 7 Tesla whole-brain fMRI experiment to asses functional activity during both anticipation (a future-oriented emotional state typically associated with worry, apprehension and freezing behaviours) and a breathlessness stimulus. We then extended this analysis beyond simple co-activation, to measure the 'functional connectivity' between the task-activated seed regions of the PAG and remote areas of the cortex, both at rest and during the anticipation and breathlessness tasks.

'Functional connectivity' within neuroimaging is a measure of the temporal synchronicity of activity within structures across the brain, and is thought to be related to the temporal coherence of neuronal activity in anatomically distinct regions (*Gerstein and Perkel, 1969*; *van den Heuvel et al., 2010*). Therefore, the term 'functional connectivity' in this paper was defined as the co-activation of the timeseries between the PAG columns and the cortex. Measures of this connectivity both at rest and during a specific task can be used to investigate the task-specific changes in functional connectivity between regions, but do not infer directionality or causality. Importantly, human investigation allows us to directly assess how functional activity and connectivity are related to subjective emotions such as the anxiety associated with threatening stimuli, while animal models rely on inferences of these perceptions from physiology such as changes in respiratory rate and heart rate. The findings of this study will help us to understand the how the different PAG columns interact with the wider cortex in humans, both at rest and when perceiving a threatening stimulus such as breathlessness.

## Results

Forty healthy, right-handed individuals were trained using an aversive delay-conditioning paradigm to associate simple shapes with either a breathlessness stimulus in the form of inspiratory resistive loading (100% contingency pairing) or no loading (0% contingency pairing). A block paradigm was used where the breathlessness shape was presented on a screen initially without the stimulus (anticipation period), followed by the addition of the resistive loading (breathlessness), and interspersed with rest periods (fixation cross, during which time there was no respiratory loading) and 'no loading' blocks (indicated by the 'no loading' shape). The following day this breathing paradigm was repeated during task fMRI scanning, with an additional resting state fMRI scan and structural scanning. Task fMRI data was analysed for mean changes in BOLD activity, and functional activity identified in both the vlPAG and lPAG was then subsequently analysed for functional connectivity with the wider cortex both at rest (using the resting state scan), and during anticipation and breathlessness (using the task fMRI scan).

### Physiology and psychology of breathlessness

Mean intensity and anxiety scores were significantly greater for breathlessness compared to unloaded breathing conditions using a paired, two-tailed t-test (intensity (±SD): 46.5 (16.0)% vs. 2.8 (3.5)%, $p<0.001$; anxiety: (±SD) 34.0 (18.8)% vs. 2.5 (3.9)%, $p<0.001$). Physiological variables for

unloaded breathing, anticipation of breathlessness and during the breathlessness stimulus are given in *Supplementary file 1A*.

## PAG task fMRI analysis

Consistent with our previous findings (*Faull et al., 2016b*), significantly increased mean BOLD activity was seen in the vlPAG (right) during anticipation of breathlessness, with an additional small cluster seen in the lPAG (left) (*Figure 1*). While no mean BOLD activity was observed in the PAG when averaged across the group during breathlessness, a significant correlation was found between lPAG and subjective scores of breathlessness intensity (*Figure 2*), consistent with the location of the lPAG activity observed during anticipation. No PAG activity was found to scale with anxiety during either anticipation or breathlessness.

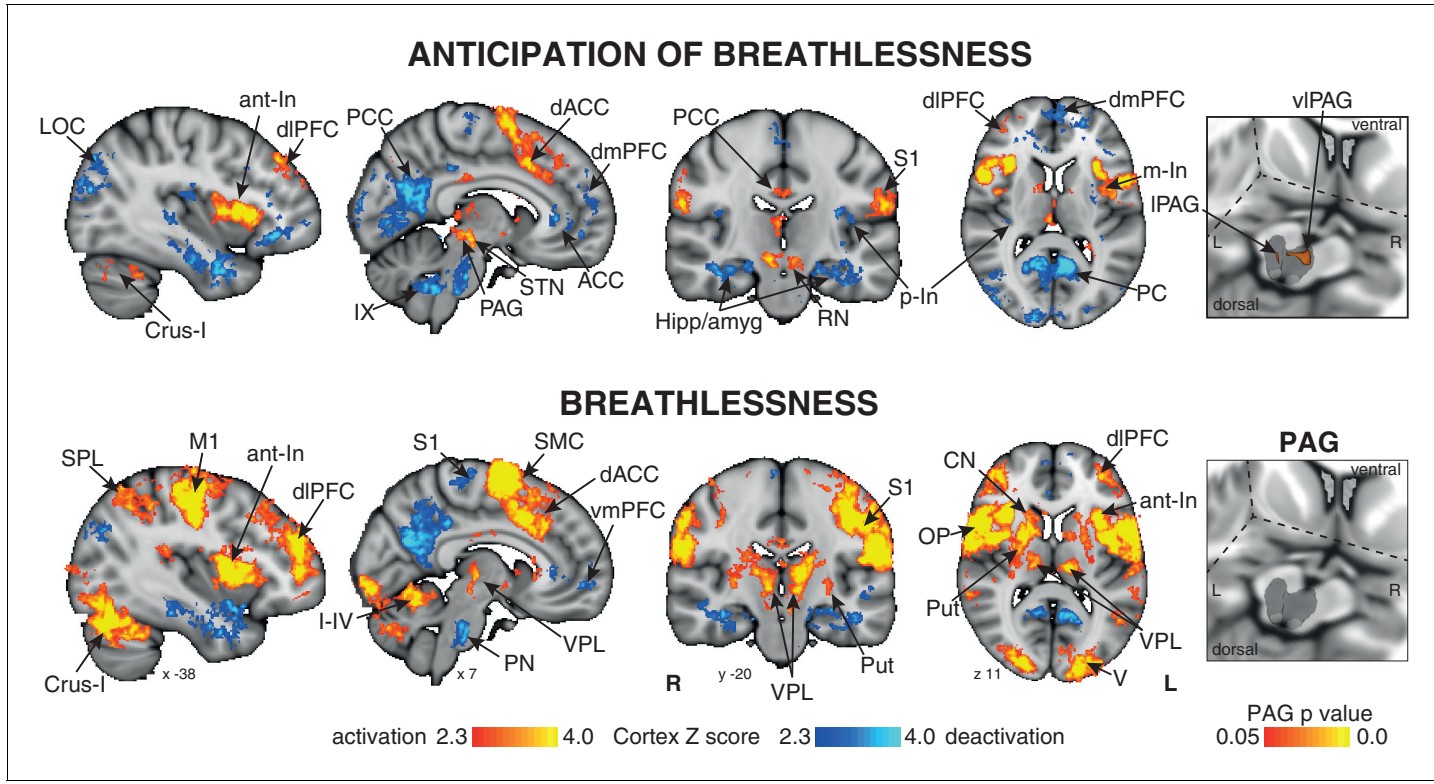

**Figure 1.** Mean BOLD response to breathlessness and anticipation of breathlessness. Right: 3-dimensional projection of the periaqueductal gray (PAG) within a cut-out of the midbrain. The images consist of a colour-rendered statistical map superimposed on a standard (MNI 1 x 1 × 1 mm) brain, and significant regions are displayed with a threshold Z > 2.3, with a cluster probability threshold of p<0.05 (corrected for multiple comparisons). Right: The grey region represents the periaqueductal gray, with significant clusters overlaid (p<0.05; non-parametric statistics, small volume-corrected for multiple comparisons using represented PAG mask). Abbreviations: PAG, periaqueductal gray; vlPAG and lPAG, ventrolateral and lateral PAG; M1, primary motor cortex; S1, primary sensory cortex; CN, caudate nucleus; SMC, supplementary motor cortex; Put, putamen; ACC, anterior cingulate cortex; dACC, dorsal anterior cingulate cortex; PCC, posterior cingulate cortex; PC, precuneus; dlPFC, dorsolateral prefrontal cortex; dmPFC, dorsomedial prefrontal cortex; vmPFC, ventromedial prefrontal cortex; Hipp, hippocampus; amyg, amygdala; a-In, anterior insula; m-In, middle insula; p-In, posterior insula; LOC, lateral occipital cortex; SPL, superior parietal lobule; STN, subthalamic nucleus; RN, Red nucleus; OP, operculum; V1, primary visual cortex; IX, I-IV and Crus-I, cerebellar lobes; thalamic nuclei: VPL, ventral posterolateral nucleus; VIN; activation, increase in BOLD signal; deactivation, decrease in BOLD signal.

The following figure supplement is available for figure 1:

**Figure supplement 1.** Task subject-level general linear model.

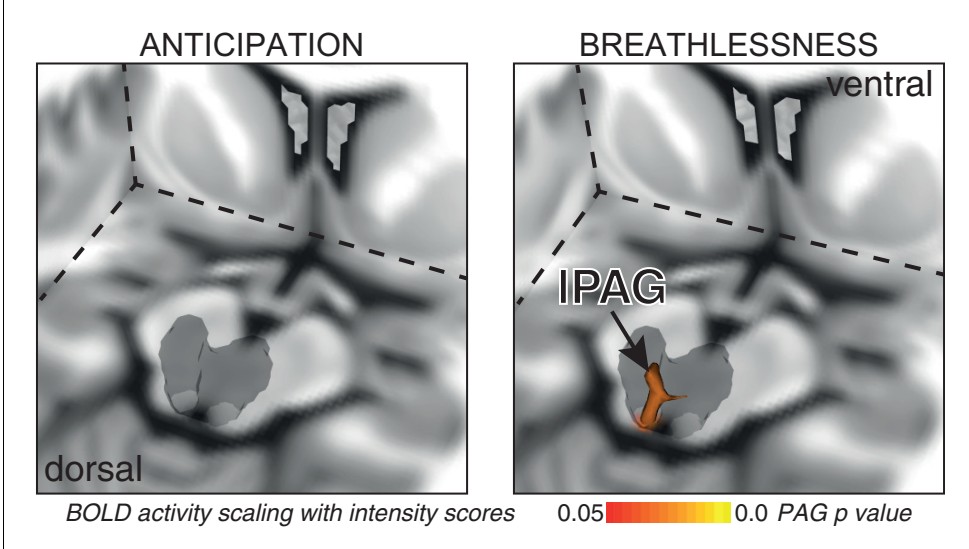

**Figure 2.** BOLD response to breathlessness that scales with intensity values across subjects. The periaqueductal gray (PAG) is represented as a 3-dimensional projection within a cut-out of the midbrain. The images consist of a 3-dimensional colour-rendered statistical map superimposed on a standard (MNI 1 x 1 × 1 mm) brain. The grey region represents the periaqueductal gray, with significant clusters overlaid (p<0.05; non-parametric statistics with threshold-free cluster enhancement, small volume-corrected for multiple comparisons using represented PAG mask).

## Cortical and subcortical task fMRI analysis

### Anticipation of breathlessness

Significantly increased BOLD activity was seen in the dorsolateral prefrontal cortex, supplementary motor cortex, middle and posterior cingulate cingulate cortices, anterior and middle insula, subthalamic nucleus, operculum, cerebellar I-IV, primary visual cortex and primary sensory cortex. Decreased BOLD activity was observed in the anterior and posterior cingulate cortices, ventromedial prefrontal cortex, dorsomedial prefrontal cortex, posterior insula, inferior precuneus, hippocampus and amygdala, primary sensory cortex, pontine nuclei, ventral inferior nuclei of the thalamus and crus IX of the cerebellum (*Figure 1*).

### Breathlessness

During breathlessness, significantly increased BOLD activity was seen in the dorsolateral prefrontal cortex, ventral posterolateral nucleus of the thalamus, putamen, caudate nucleus, primary sensory and motor cortices, supplementary motor cortex, middle cingulate cortex, anterior and middle insula, subthalamic nucleus, operculum, cerebellar I-IV, primary visual cortex and primary sensory cortex. Decreased BOLD activity was seen in the anterior and posterior cingulate cortices, ventromedial prefrontal cortex, inferior precuneus, hippocampus and amygdala, primary sensory cortex, and pontine nuclei (*Figure 1*).

## Resting functional connectivity of the active PAG columns

A 3-dimensional seed region (seed radius 2 mm) was taken in the center of the significant activity found in both the vlPAG (seed location MNI space: x 7; y −29; z −8) and lPAG (seed location MNI space: x −3; y −32; x −8). These vlPAG and lPAG seeds were used to create a mean time series of the resting BOLD data in each subject, and the significant functional connectivity to the rest of the brain was found using a general linear regression of this timecourse at rest. The vlPAG seed showed significant functional connectivity to the right dorsomedial and dorsolateral prefrontal cortex, anterior cingulate cortex, anterior/middle insula and amygdala, and bilateral connectivity to the ventral posterolateral, ventrolateral and dorsomedial nuclei of the thalamus, primary visual cortex, posterior

cingulate cortex, precuneus, hippocampus and parahippocampal gyri, subthalamic nucleus and I-IV, V, VI, crus-I lobes of the cerebellum (*Figure 3*). Conversely, the lPAG seed displayed significant resting connectivity to the right primary motor and sensory cortices, superior temporal gyrus, caudate nucleus and bilateral connectivity to the putamen, hippocampus and parahippocampal gyri, and ventral posteromedial thalamic nuclei (*Figure 3*).

## Functional task (PPI) connectivity of the active PAG columns

A psychophysiological interaction (PPI) analysis (*Friston et al., 1997*; *O'Reilly et al., 2012*) was conducted for each of the active PAG columns during both anticipation and breathlessness, using the same seed regions in the vlPAG and lPAG.

### Anticipation of breathlessness
The connectivity of vlPAG to the bilateral primary motor and sensory cortices, right supplementary motor cortex, left supramarginal gyrus, and bilateral cerebellar I-IV, V was found to negatively scale with intensity scores across subjects (*Figure 4*).

### Breathlessness
The connectivity of the vlPAG to the bilateral inferior precuneus, right putamen, and bilateral cerebellar I-IV, V was again found to negatively correlate with intensity scores. Additionally, the connectivity of the vlPAG to the left middle/inferior insula and right parahippocampal gyrus positively scaled with anxiety scores (*Figure 4*). The connectivity of the lPAG to the left amygdala and anterior

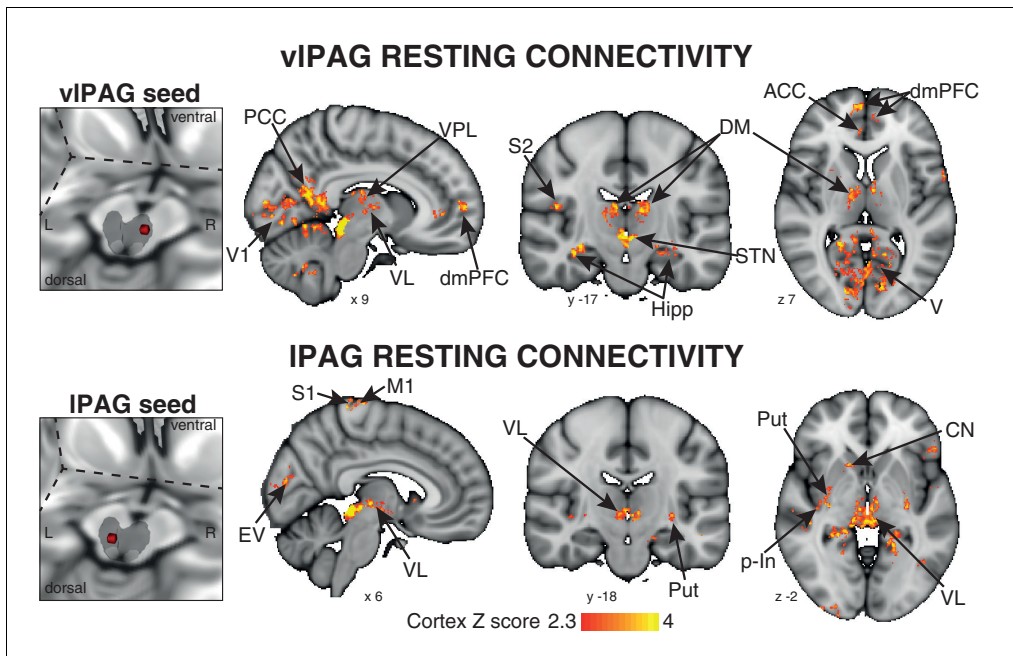

**Figure 3.** Mean resting state functional connectivity of the active PAG seed regions. Left: 3-dimensional projection of the periaqueductal gray (PAG) within a cut-out of the midbrain, and the seed placement within the PAG according to functional BOLD activity results. Right: Resting functional connectivity (correlation strength) between the PAG seed and the rest of the brain. The images consist of a colour-rendered statistical map superimposed on a standard (MNI 1 x 1 × 1 mm) brain, and significant regions are displayed with a threshold Z > 2.3, with a cluster probability threshold of p<0.05 (corrected for multiple comparisons). Abbreviations: PAG, periaqueductal gray; vlPAG and lPAG, ventrolateral and lateral PAG; M1, primary motor cortex; S1, primary sensory cortex; CN, caudate nucleus; ACC, anterior cingulate cortex; PCC, posterior cingulate cortex; p-In, posterior insula; dmPFC, dorsomedial prefrontal cortex; Hipp hippocampus; V1, primary visual cortex; V, visual cortex; thalamic nuclei: VP, ventral posterolateral nucleus; VL, ventrolateral nucleus; DM, dorsomedial nucleus.

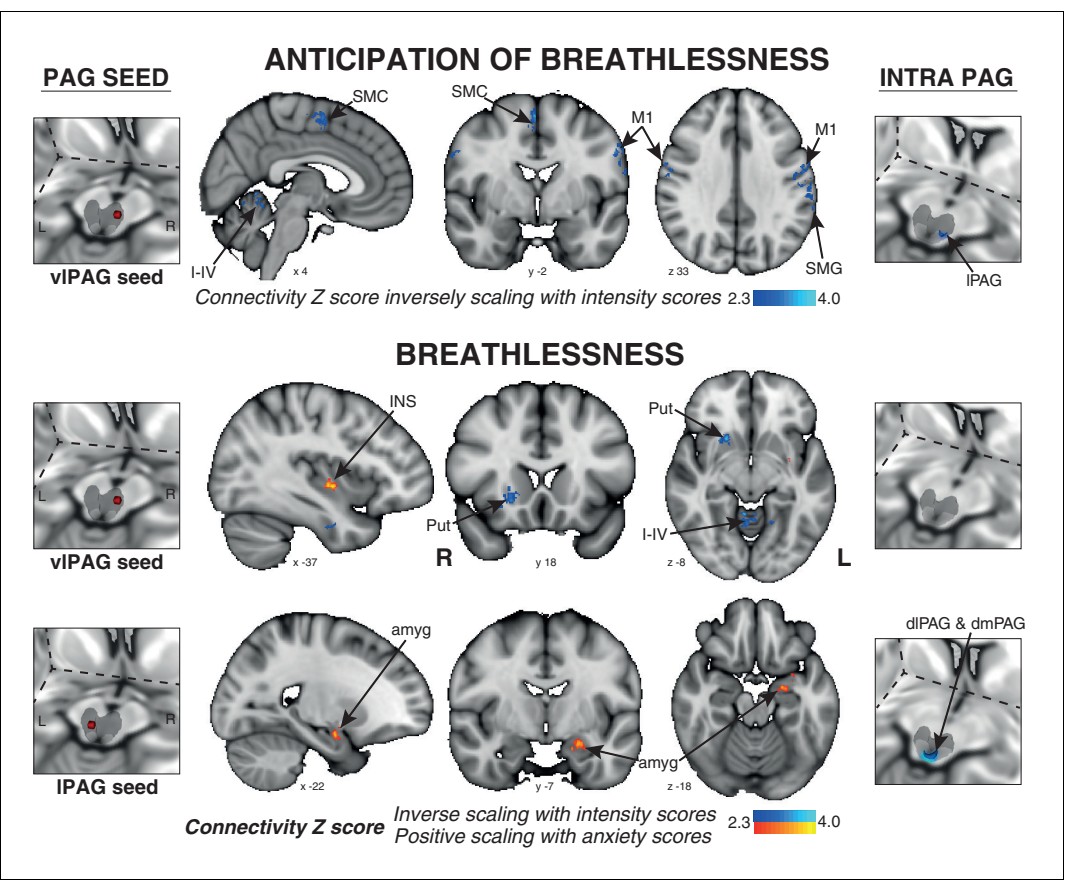

**Figure 4.** Psychophysiological interactions (PPI) of the active PAG seed regions that scale with subjective ratings of intensity and anxiety of breathlessness across subjects. Left: 3-dimensional projection of the periaqueductal gray (PAG) within a cut-out of the midbrain, and the seed placement within the PAG according to functional BOLD activity results. Middle: Task functional connectivity (using psychophysiological interaction analyses; PPI) between the PAG seed and the rest of the brain during the anticipation and breathlessness tasks. The images consist of a colour-rendered statistical map superimposed on a standard (MNI 1 x 1 × 1 mm) brain, and significant regions are displayed with a threshold Z > 2.3, with a cluster probability threshold of p<0.05 (corrected for multiple comparisons). Right: Intra-PAG connectivity measured using permutation-testing within the whole PAG. Abbreviations: PAG, periaqueductal gray; vlPAG and lPAG, ventrolateral and lateral PAG; M1, primary motor cortex; SMC, supplementary motor cortex; Put, putamen; amyg, amygdala; INS, middle insula; I-IV, cerebellar lobe.

hippocampus also positively scaled with anxiety scores (*Figure 4*), while the intra-PAG connectivity to the dlPAG and dmPAG inversely scaled with breathlessness intensity scores (*Figure 4*).

## Discussion

In this study we have demonstrated vlPAG and lPAG activity during conditioned anticipation of breathlessness, and lPAG activity during breathlessness that scaled with subjective scores of breathlessness intensity. We have revealed resting functional connectivity profiles of these active PAG columns with the wider cortex that align with current theories of threat-induced behaviors, where the lPAG is functionally connected to sensorimotor structures in preparation for fight/flight, whilst the vlPAG is functionally connected to affective and evaluative prefrontal areas. Furthermore, during anticipation of breathlessness, the vlPAG demonstrated reduced connectivity to both the lPAG and sensorimotor structures, consistent with a potential 'freeze' response.

## lPAG in the fight/flight response to threat

The lPAG (alongside the dlPAG) has long been hypothesised to act within the behavioural response to an escapable stressor for 'fight' or 'flight' responses (*Bandler et al., 2000*; *Bandler and Shipley, 1994*; *Carrive, 1993*; *Keay and Bandler, 2001*). Our resting connectivity analysis revealed communications between the lPAG and sensorimotor structures such as primary motor and sensory cortices, caudate and putamen, consistent with known white matter connectivity profiles determined in humans using diffusion tractography (*Ezra et al., 2015*). This profile demonstrates plausible pathways for involvement in active responses through functional sensorimotor connections to the primary motor and sensory cortices and caudate nucleus (*Cools, 1980*). The lPAG showed no changes in connectivity profile during anticipation, while during breathlessness exhibited increased connectivity to the amygdala correlating with subjective anxiety scores. These results reveal a potential anxiety-encoding pathway through the amygdala during the perception of the breathlessness stimulus (*Davis, 1992*; *Stein et al., 2007*). This pathway may incorporate into both existing neurocognitive anxiety models of prefrontal-amygdala top-down contributions to threat responses (*Bishop, 2009*, *2007*), and amygdala - bed nucleus of the stria terminalis triangulation with the PAG previously shown in animals (*Gray and Magnuson, 1992*; *Hopkins and Holstege, 1978*). Therefore, in this study we have demonstrated not only lPAG activity during breathlessness, but functional connectivity of the lPAG that may potentiate active responses to an escapable stressor and the anxiety encoding of this threat. We do not yet know whether these findings are specific to respiratory threat or generalizable. Therefore, future work investigating functional connectivity of the substructures of the PAG could employ alternative threatening stimuli, for example pain, to extend our understanding of the role of the PAG columns across generalized threat responses.

## vlPAG in the 'freeze' response to threat

Conversely, the vlPAG has been hypothesised to act within the behavioural response to an inescapable stressor in animals, exhibiting physiological changes associated with 'freezing' behaviours. In the current study, the conditioned anticipation of a resistive load represents an inescapable respiratory threat; a human threat analogue to experiments demonstrating freezing behaviours to conditioned anticipation of electric shock in animals (*De Oca et al., 1998*), which are attenuated by lesioning the vlPAG. To extend our understanding of the role of the vlPAG in these freezing behaviours, in this experiment we sought to understand how the vlPAG might interact with the wider cortex and subcortex within this response in humans.

Resting connectivity of the vlPAG demonstrated communication with evaluative and affective areas such as the dorsolateral and dorsomedial prefrontal cortices (*Gusnard et al., 2001*; *MacDonald et al., 2000*; *Miller and Cohen, 2001*), posterior cingulate cortex (*Vogt et al., 1992*), amygdala and insula (*Stein et al., 2007*). With the acquisition and extinction of learned fear responses necessitated by the amygdala (*Maren and Fanselow, 1996*), and prefrontal cortical connections regulating amygdala-mediated responses to previously-conditioned stimuli (*Rosenkranz et al., 2003*), these connections would be of utmost importance to elicit appropriate freezing behaviours in conditioned fear responses. However, this resting connectivity profile of the vlPAG displays no functional connections to primary motor structures through which it might orchestrate a 'freeze' response to an inescapable threat. Therefore, to instigate freezing during anticipation of breathlessness, our exploratory analyses have shown that the vlPAG appears to *disconnect* with both the lPAG and primary motor, sensory and cerebellar structures in those subjects who perceived greater breathlessness intensity, and increased connectivity to the insula with greater anxiety. Lastly, during breathlessness the vlPAG also demonstrated reduced connectivity to motor structures (limited to the putamen and cerebellum) and increased to the anxiety-processing inferior middle insula (*Liotti et al., 2001*; *Simmons et al., 2006*), suggesting that this stimulus may also be eliciting an element of perceived inescapable threat, with subjects secured and immobile on a breathing system in the scanner.

## Intra-PAG connectivity and interaction between fight/flight and freeze responses

While the vlPAG and lPAG appear to have very different functions within the threat response to breathlessness, it is possible that inter-columnar communications may allow appropriate stimulus

encoding and subsequent behavioural responses. All PAG columns have been shown to project to all other columns in animals (*Jansen et al., 1998*), and vlPAG and lPAG outputs have also demonstrated reciprocal inhibition to each other and downstream targets (*Carrive, 1993*; *Jansen et al., 1998*; *Lovick, 1992*; *Tovote et al., 2016*). While the translation to humans is not yet known, a similar system would allow intra-PAG communications (such as those demonstrated in *Figure 4* during anticipation and breathlessness) that may permit switching or selecting between threat behaviours. Therefore, the apparent reduced connectivity to motor structures during 'freeze' may be via inhibition of the lPAG and its downstream connections, inhibiting the 'fight/flight' to instigate freezing.

## Respiratory evidence from animal models

For the specific respiratory response to a breathlessness threat, we can draw comparisons between the findings from this study and previous respiratory investigations in animals. Firstly, animal models have shown direct stimulation of the lPAG to result in tachypnea in the rat (*Subramanian and Holstege, 2013*) and cat (*Subramanian and Holstege, 2009*; *Zhang et al., 1994*), and have even postulated the presence of a suffocation alarm within PAG columns including the lPAG (*Lopes et al., 2012*; *Schimitel et al., 2012*). Here we see lPAG activity in response to a respiratory threat that parallels subjective intensity, where active increases in breathing effort are required to overcome the inspiratory resistance. This synergy between animal and human models allows us to develop our understanding of the potential role of the lPAG in overcoming perceived *respiratory* threats such as airway obstruction and breathlessness, as well as a possible more generalised tachypneic response to wider threat perceptions.

Whilst the vlPAG is considered an integral part of the global freezing response to a conditioned threat, the respiratory response during freezing is less well understood. Rather than measure breathing changes to a conditioned threat, animal models have instead employed direct PAG stimulation and subsequently measured the breathing responses, reporting changes such as irregular breathing (*Subramanian et al., 2008*), expiratory prolongation (*Subramanian, 2013*), and apneas (*Subramanian and Holstege, 2013*). Furthermore, no animal studies to date have employed a conditioned *respiratory* threat to evoke defensive behaviour, and in the current study in humans we find a small increase in ventilation during anticipation of respiratory threat (Supplemental Table 1), alongside vlPAG and lPAG activity. It is possible that the respiratory response to an upcoming conditioned threat is multi-faceted, and while vlPAG activity may inhibit respiratory control centres (*Subramanian and Holstege, 2013*), the counterpart activity observed in the lPAG may override this and result in stimulation of respiratory control centres (*Subramanian and Holstege, 2013*; *Zhang et al., 1994*). Therefore, it is clear that further investigation is needed to understand the intricate control of respiration by the PAG in the face of a both conditioned respiratory threat such as breathlessness, and towards other conditioned threats.

## Breathlessness in disease

This division in function and connections between the columns that make up the substructure of PAG within the wider system of breathlessness perception may prove to be of great clinical importance. Conditions such as chronic obstructive pulmonary disease (COPD), asthma and panic disorder are often characterised by misperceptions of respiratory sensations and inappropriate behavioural responses (*Janssens et al., 2009*; *Magadle et al., 2002*; *Wisnivesky et al., 2010*). We can now begin to provide a platform for investigating related adaptations of the PAG and its functional connections to remote brain regions in these diseases, in addition to previously identified areas such as the medial prefrontal cortex (*Herigstad et al., 2015*; *Pattinson, 2015*; *Pattinson and Johnson, 2014*). It is possible that heightened activity and 'sensitisation' of the lPAG may result in greater fearful encoding of breathlessness, resulting in an exacerbated anticipatory threat response (potentiated by the vlPAG) to breathlessness stimuli, such as the avoidance of daily activities. It appears that the communication between the columns of the PAG and the wider cortex may be focal in orchestrating human behavioural responses to perceived threat, marking a key step towards understanding and intervening in those with misplaced threat responses, such as the heightened anxiety often observed with COPD (*Janssens et al., 2011*), asthma (*Banzett et al., 2000*; *Janssens et al., 2009*), panic disorder (*Smoller et al., 1996*) or pain (*Zambreanu et al., 2005*).

## Limitations

This study has used 7 Tesla fMRI to investigate differential activity and functional connectivity of the human PAG columns in association with perceived threat, and in this instance the respiratory threat of breathlessness. Whilst the use of ultra-high field scanners affords us higher-resolutions that facilitate exploration of the subdivisions of the human PAG, there are some constraints within fMRI scanning and analysis that need to be addressed. First and foremost, all fMRI scanning involves imposing a constrained behavioural or cognitive task and measuring the associated brain signal, rather than direct stimulation of an area of interest and measurement of resulting physiology or behaviour (as often employed in animal models). Therefore, when using fMRI, an area of interest (such as an individual column of the PAG) cannot be viewed in isolation, nor its activity identified as causal to the outcome of the task. Additionally, the physical constraints imposed by an MRI scanner, both physical (cylindrical) constraints and the possible effect of large movements on the magnetic field that may corrupt signal, somewhat limit the tasks and the repertoire of available behaviours from the subject, who must remain somewhat motionless during experimental sessions.

Interestingly, in both this study and the original investigation (*Faull et al., 2016b*) we did not identify dorsal PAG activity during any of our tasks. The dorsal PAG has previously been implicated in sensory processing and active defence reactions, and is thought to be a major centre in the response to an escapable threat (*Bandler et al., 2000*; *Bandler and Carrive, 1988*; *Canteras and Goto, 1999*; *Dampney et al., 2013*; *Keay and Bandler, 2001*; *Lopes et al., 2014*). It is possible that the specific breathlessness paradigm we employed, and the enforced movement constraints of the MRI scanner may have influenced these results and limited dorsal PAG activity. This experimental situation differs vastly from many studies in freely moving animals with implanted electrodes or lesions (*Bandler and Carrive, 1988*; *Canteras and Goto, 1999*; *Carrive, 1993*; *Carrive and Bandler, 1991*), or direct-stimulation studies of PAG columns in decerebrate animals (*Subramanian et al., 2008*). Whilst there have been reports of dorsal PAG activity in humans using functional MRI and threatening word presentations (*Satpute et al., 2013*), the functional subdivision of the PAG did not follow the consensus of PAG subdivisions from animal literature, and incorporated the separate, well-differentiated nuclei on the ventral border of the PAG within its mask. In the present study we have attempted to functionally localise human PAG activity based on the extensive animal models, and recent evidence in humans (*Ezra et al., 2015*).

We did, however, observe differences in functional connectivity between lPAG and dorsal PAG during breathlessness, which inversely correlated with behavioural scores of subjective intensity (*Figure 4*). Therefore, it is possible that intra-PAG functional connections are important within the sensory processing of breathlessness sensations. Alternatively, these findings may represent interspecies differences in PAG activity and/or connectivity when responding to the threat of breathlessness. Therefore, whilst we were unable to identify dorsal PAG activity within our specific experimental task, further work is required to more fully elucidate the roles of the different PAG subdivisions across a range of threatening stimuli in humans.

This work attempts to quantify 'functional connectivity' between the active PAG columns and cortex, both at rest and during breathlessness anticipation and perception. The functional connectivity methods employed in this study (using correlation strength between seed areas of interest and the wider brain) measure time-locked associations between brain regions (*Gerstein and Perkel, 1969*; *van den Heuvel et al., 2010*), but importantly do not measure causality. Therefore, we are currently unsure of the driving, causal centres within these functionally-associated areas, and may be insensitive to functional associations that are too temporally asynchronous or dynamic for our statistical thresholds to identify. The temporal blurring of BOLD signal due to the HRF, coupled with the long (3 s) volume repetition time (TR) employed in this study would mean vastly greater statistical power would be likely necessary to attempt either intricate causality modelling ('effective connectivity') using methods such as dynamic causal modelling (DCM) (*Friston et al., 2003*), or more dynamic versions of functional connectivity (*Chang and Glover, 2010*). Whilst techniques such as simultaneous multi-slice imaging are now becoming more mainstream (*Feinberg et al., 2010*; *Moeller et al., 2010*), researchers may need to be wary of potential blurring between slices deep in the brain within image reconstruction of these acquisitions (*Feinberg et al., 2010*), which may temper our ability to functionally isolate structures such as the subdivisions of the PAG.

Finally, the use of a respiratory stimulus within fMRI requires careful consideration, and is discussed in more detail in *Hayen et al. (2017)*. Physiological noise can be induced by both movement during the respiratory cycle, and bulk-susceptibility issues where the change in volume of air in the chest results in distortions of the magnetic field (*Brooks et al., 2013*; *Glover et al., 2000*; *Harvey et al., 2008*; *Raj et al., 2001*). Therefore, brain signal that is directly linked with these periodic fluctuations cannot be disentangled from physiological noise, and will be regressed out when using rigorous noise correction procedures. In the present study we were not attempting to directly link PAG activity to changes in respiratory rhythm generating centres, instead the primary aim was to link PAG activity with the perception of a respiratory threat. Future research looking to identify PAG-related changes in human respiratory rhythm generation will need to carefully consider possible experimental approaches to achieve this.

## Conclusions

In this study we have identified the PAG as a potential hub within the orchestration of the human behavioural responses to threat. We have found that the lPAG is active during both anticipation of breathlessness and during the sensation of breathlessness. The lPAG is functionally connected to sensorimotor cortical structures at rest for possible active fight/flight behavioural pathways, and during breathlessness it exhibits increased connectivity to the amygdala in that correlates with subjective anxiety scores. Conversely, the vlPAG is active only during anticipation of breathlessness, is functionally connected to fronto-limbic evaluative structures at rest, and during anticipation *disconnects* from the lPAG and sensorimotor structures to possibly potentiate a 'freezing' response to an inescapable threat. This work extends our previous findings of differential vlPAG and lPAG activity during breathlessness, and is a first step in understanding how these PAG columns may interact both with each other and the wider cortex in the generation of behavioural responses to threat in humans.

## Materials and methods

### Subjects

Subjects studied consisted of 40 healthy, right-handed individuals (20 males, 20 females; mean age ± SD, 26 ± 7 years), with no history of smoking or any respiratory disease. Within this cohort there were two groups; 20 subjects who regularly participated in endurance sport and 20 age- and sex-matched (±2 years) sedentary subjects. Prior to scanning, all subjects underwent physiological breathlessness testing during exercise and chemostimulated hyperpnea, which have been presented elsewhere (*Faull et al., 2016a*). No differences in either inspiratory mouth pressure or subjective breathlessness ratings were seen between athletes and sedentary subjects for any task conditions (*Supplementary file 1B*), and subsequently no BOLD PAG differences were observed between groups for any task. Therefore, a group difference regressor was included in the analysis but will not be considered as part of this manuscript.

### Stimuli and tasks

Subjects were trained using an aversive delay-conditioning paradigm to associate simple shapes with an upcoming breathlessness (inspiratory resistance) stimulus. Two conditions were trained: (1) A shape that always predicted upcoming breathlessness (100% contingency pairing), and (2) A shape that always predicted unloaded breathing (0% contingency pairing with inspiratory resistance). These two conditions were identical to our previous work (*Faull et al., 2016b*), however in this instance no 'uncertain' condition (50% contingency pairing) was used.

The 'certain upcoming breathlessness' symbol was presented on the screen for 30 s, which included a varying 5–15 s anticipation period before the loading was applied. The 'unloaded breathing' symbol was presented for 20 s, and each condition was repeated 14 times in a semi-randomised order. A finger opposition task was also included in the protocol, where an opposition movement was conducted between the right thumb and fingers, with the cue 'TAP' presented for 15 s (10 repeats). Finger opposition was used as a control task (*Faull et al., 2016b, 2015*; *Pattinson et al., 2009a*); results are not presented here. Conscious associations between cue and threat level (cue

contingencies) were required and verified in all subjects by reporting the meaning of each of the symbols following the training session and immediately prior to the MRI scan.

Rating scores of breathing difficulty were recorded after every symbol and at the beginning and end of the task, using a visual-analogue scale (VAS) with a sliding bar to answer the question 'How difficult was the previous stimulus?' where the subjects moved between 'Not at all difficult' (0%) and 'Extremely difficult' (100%). Subjects were also asked to rate how anxious each of the symbols made them feel ('How anxious does this symbol make you feel?') using a VAS between 'Not at all anxious' (0%) and 'Extremely anxious' (100%) immediately following the functional MRI protocol.

## Breathing system and physiological measurements

A breathing system was used to remotely administer periods of inspiratory resistive loading to induce breathlessness as predicted by the conditioned cues (*Hayen et al., 2013a*, *2015*), identical to that previously described (*Faull et al., 2016b*). End-tidal oxygen and carbon dioxide were maintained constant (*Faull et al., 2016b*). The subject's nose was blocked using foam earplugs and they were asked to breathe through their mouth for the duration of the experiment. Physiological measures were recorded continuously during the training session and MRI scan as previously described (*Faull et al., 2016b*).

## MRI scanning sequences

MRI was performed with a 7 T Siemens Magnetom scanner, with 70 mT/m gradient strength and a 32 channel Rx, single channel birdcage Tx head coil (Nova Medical).

### Blood Oxygen Level Dependent (BOLD) scanning

A T2*-weighted, gradient echo planar image (EPI) was used for functional scanning. The field of view (FOV) covered the whole brain and comprised 63 slices (sequence parameters: echo time (TE), 24 ms; repetition time (TR), 3 s; flip angle, 90°; voxel size, $2 \times 2 \times 2$ mm; FOV, 220 mm; GRAPPA factor, 3; echo spacing, 0.57 ms; slice acquisition order, descending), with 550 volumes (scan duration, 27 min 30 s) for the task fMRI, and 190 volumes (scan duration, 9 min 30 s) for the resting-state acquisition (eyes open).

### Structural scanning

A T1-weighted structural scan (MPRAGE: magnetisation-prepared rapid gradient-echo), sequence parameters: TE, 2.96 ms; TR, 2200 ms; flip angle, 7°; voxel size, $0.7 \times 0.7 \times 0.7$ mm; FOV, 224 mm; inversion time, 1050 ms; bandwidth; 240 Hz/Px) was acquired. This scan was used for registration of functional images.

### Additional scanning

Fieldmap scans (sequence parameters: TE1, 4.08 ms; TE2, 5.1 ms; TR, 620 ms; flip angle, 39°; voxel size, $2 \times 2 \times 2$ mm) of the $B_0$ field were also acquired to assist distortion-correction.

## Analysis

### Preprocessing

Image processing was performed using the Oxford Centre for Functional Magnetic Resonance Imaging of the Brain Software Library (FMRIB, Oxford, UK; FSL version 5.0.8; http://www.fmrib.ox.ac.uk/fsl/). The following preprocessing methods were used prior to statistical analysis: motion correction and motion parameter recording (using MCFLIRT: Motion Correction using FMRIB's Linear Image Registration Tool [*Jenkinson et al., 2002*]), removal of the non-brain structures (skull and surrounding tissue) (using BET: Brain Extraction Tool [*Smith, 2002*]), spatial smoothing using a full-width half-maximum Gaussian kernel of 2 mm, and high-pass temporal filtering (Gaussian-weighted least-squares straight line fitting; 120 s). Spatial smoothing was limited to 2 mm, to maintain the improved functional resolution ($2 \times 2 \times 2$ mm) afforded by 7 Tesla imaging. Minimal spatial smoothing, whilst conservative, reduces blurring between small structures, such as the columns of the PAG. Distortion correction of EPI scans was conducted using a combination of FUGUE (FMRIB's Utility for Geometrically Unwarping EPIs [*Holland et al., 2010*; *Jenkinson, 2001*; *Jezzard, 2012*]) and BBR (Boundary Based Registration; part of the FMRI Expert Analysis Tool, FEAT, version 6.0 [*Greve and Fischl,*

*2009*]). Distortion correction is particularly important in areas of the brain that suffer considerable distortions due nearby bony interfaces and proximity to sinuses, such as occurs in the brainstem. Further information on preprocessing can be found in the *Supplementary file 2*.

Due to the location of the PAG and its susceptibility to physiological noise (*Brooks et al., 2013*), rigorous data denoising was conducted using a combination of independent component analysis (ICA) and retrospective image correction (RETROICOR [*Brooks et al., 2013*; *Harvey et al., 2008*]), as previously described (*Faull et al., 2016b*; *Hayen et al., 2017*). Briefly, this process first involved decomposing the data using automatic dimensionality estimation (*Kelly et al., 2010*). Head motion regressors calculated from the motion correction preprocessing step (using MCFLIRT in this case) were regressed out of the data alongside the 'noise' components identified during ICA denoising, during preprocessing of the functional scans prior to first level analysis. Physiological recordings of heart rate and respiration from respiratory bellows were transformed into regressors (three cardiac, four respiratory harmonics, an interaction term and a measure of respiratory volume per unit of time (RVT) [*Harvey et al., 2008*]) corresponding to each acquisition slice, and the signal associated with this noise was isolated using linear regression, adjusted for any interaction with previously-identified ICA noise components and then subtracted from the data.

## Image registration

After preprocessing, the functional scans were registered to the MNI152 (1 x 1 × 1 mm) standard space (average T1 brain image constructed from 152 normal subjects at the Montreal Neurological Institute (MNI), Montreal, QC, Canada) using a two-step process: (1) Registration of subjects' whole-brain EPI to T1 structural image was conducted using BBR (6 DOF) with (nonlinear) fieldmap distortion-correction (*Greve and Fischl, 2009*), and (2) Registration of the subjects' T1 structural scan to 1 mm standard space was performed using an affine transformation followed by nonlinear registration (using FNIRT: FMRIB's Non-linear Registration Tool [*Andersson et al., 2007*]).

## Functional voxelwise and group analysis

Functional data processing was performed using FEAT (FMRI Expert Analysis Tool, part of FSL). The first-level analysis in FEAT incorporated a general linear model (*Woolrich et al., 2004*), with the following regressors: Breathlessness periods (calculated from physiological pressure trace as onset to termination of each application of resistance); anticipation of breathlessness (calculated from onset of anticipation symbol to onset of resistance application); unloaded breathing (onset and duration of 'unloaded breathing' symbol); and finger opposition (onset and duration of finger opposition screen instruction). Additional regressors to account for relief from breathlessness, periods of rating using the button box, demeaned ratings of intensity between trials, and a period of no loading following the final anticipation period (for decorrelation between anticipation and breathlessness) were also included in the analysis. A final partial pressure of end-tidal carbon dioxide ($P_{ET}CO_2$) regressor was formed by linearly extrapolating between end-tidal carbon dioxide ($CO_2$) peaks, and included in the general linear model to decorrelate any $P_{ET}CO_2$-induced changes in BOLD signal from the respiratory tasks (*Faull et al., 2016b*, *2015*; *McKay et al., 2008*; *Pattinson et al., 2009a*, *2009b*).

Functional voxelwise analysis incorporated HRF modeling using three optimal basis functions (FLOBS: FMRIB's Linear Optimal Basis Set), instead of the more common gamma waveform. These three regressors allow for more flexibility in the timing and shape of the haemodynamic response function (HRF), which may be caused by slice-timing delays, differences in the HRF between the brainstem and cortex, or differences between individuals (*Devonshire et al., 2012*; *Handwerker et al., 2004*). This HRF flexibility is especially important when modelling both small regions (such as the PAG) and large areas of cortex in a single model, where the HRF likely differs across these regions. Time-series statistical analysis was performed using FILM (FMRIB's Improved Linear Model), with local autocorrelation correction (*Woolrich et al., 2001*). The second and third waveforms were orthogonalised to the first to model the 'canonical' HRF, of which the parameter estimate was then passed up to the group analysis in a mixed-effects analysis using FLAME 1 + 2 (FMRIB's Local Analysis of Mixed Effects [*Woolrich et al., 2004*]). A group analysis was conducted, consisting of mean BOLD activity, demeaned breathlessness intensity and anxiety covariates to investigate BOLD activity in the PAG and cortex that scaled with these behavioural scores, and a group difference regressor (to account for any differences between athletes and controls). An

additional residual motion regressor was included in this group analysis, consisting of subject-specific DVARS values (see Supplementary material), to ensure the results were independent of any effects of residual motion.

For whole-brain results, Z statistic images were thresholded using clusters determined by Z > 2.3 and a (family-wise error (FWE) corrected) cluster significance threshold of p<0.05. Recent scrutiny of fMRI statistical methods has raised concerns over cluster-based thresholding when combined with parametric testing (*Eklund et al., 2016*). However, FILM-based autocorrelation correction in FSL minimizes inflations in the rate of false positive results, and minimal spatial smoothing (2 mm) employed in this study lessens the broadening of the data distribution tails ('heavy tails') that occur when smoothing to a more standard 6 mm, further minimizing changes in false positives (*Eklund et al., 2016*). When investigating activity within the PAG as an a-prior area of interest, a small-volume mask of the whole PAG was used and permutation testing was employed with threshold-free cluster enhancement (*Smith and Nichols, 2009*) and a (FWE-corrected) cluster significance threshold of p<0.05. Permutation testing has been shown to robustly control for appropriate false positive rates with both voxelwise and cluster thresholding (*Eklund et al., 2016*). The main contrasts of interest in this analysis were the BOLD activity that was greater during breathlessness than baseline, and the BOLD activity elicited during anticipation of breathlessness, i.e. anticipation of breathlessness > unloaded breathing.

## Resting functional connectivity analysis

Voxelwise single subject and group analyses were also performed on the acquired resting state scan. A 3-dimensional seed region (seed radius 2 mm) was taken in the center of the significant activity found in both the vlPAG and lPAG for the mean BOLD activity during anticipation of breathlessness. These vlPAG and lPAG seeds were used to create a mean time series of the resting BOLD data in each subject. The T1 structural image from each subject was segmented using hard segmentation in FAST (*Zhang et al., 2001*), transformed into functional EPI space and the mean time series from the cerebrospinal fluid and white matter were included as noise regressors of no interest. No convolution was applied, and a temporal derivative was included for each regressor to account for small delays in the connectivity between areas of the brain. A mixed-effects group analysis of the group mean, behavioural scores and residual motion was then performed on each lower level regressor. A pre-threshold grey matter mask was also applied to constrain our analysis to connectivity of the PAG seeds to the cortical and subcortical grey matter.

## Task functional connectivity analyses

Functional connectivity between the PAG and the rest of the brain during the task conditions of anticipation and breathlessness was measured using a psychophysiological interaction (PPI) analysis (*Friston et al., 1997*; *O'Reilly et al., 2012*) for both of the task-active PAG columns. This analysis examines correlated brain activity across brain regions but does not infer causality. PPI analysis involves the addition of a timeseries from a physiological region of interest (ROI), in this case a PAG seed, into the general linear model used for the task analysis, and a final 'interaction' regressor between this ROI timecourse and the task regressor of interest. This interaction regressor becomes the PPI, and is a measure of the strength of the correlation between the seed timecourse and the other voxels in the brain in the task of interest, over and above other tasks in the model.

The same seeds as those used in the resting functional connectivity analysis were used for the physiological ROI timeseries. The PPI regressor was the interaction between the PAG seed and the task regressor (either breathlessness or an anticipation of breathlessness > unloaded breathing regressor). Separate PPI analyses were conducted for the lPAG and vlPAG for their functional connectivity during both anticipation of breathlessness > unloaded breathing, and during breathlessness > baseline. Each of the resultant four PPI parameter estimates (vlPAG and lPAG connectivity during breathlessness; vlPAG and lPAG connectivity during anticipation) were taken to a middle level fixed-effects analysis for each subject, with a final mixed-effects group analysis conducted consisting of group mean, behavioural scores and residual motion on each PPI regressor. A pre-threshold grey-matter mask was applied to constrain our analysis to connectivity of the PAG seeds to the cortical and subcortical grey matter.

## Acknowledgements

This research was supported by the JABBS Foundation. This research was further supported by the National Institute for Health Research, Oxford Biomedical Research Centre based at Oxford University Hospitals NHS Trust and University of Oxford. Olivia K Faull was supported by the Commonwealth Scholarship Commission. The authors would like to thank Dr Falk Eippert, Dr Ben Ainsworth and Dr Anja Hayen for their thoughts and comments on the manuscript.

## Additional information

### Funding

| Funder | Author |
| --- | --- |
| The JABBS Foundation | Olivia K Faull<br>Kyle TS Pattinson |
| National Institute for Health Research | Kyle TS Pattinson |
| Medical Research Council | Kyle TS Pattinson |
| Commonwealth Scholarship Commission | Olivia K Faull |

The funders had no role in study design, data collection and interpretation, or the decision to submit the work for publication.

### Author contributions

OKF, Conceptualization, Data curation, Formal analysis, Funding acquisition, Investigation, Methodology, Writing—original draft, Writing—review and editing; KTSP, Conceptualization, Supervision, Funding acquisition, Investigation, Project administration, Writing—review and editing

### Author ORCIDs

Olivia K Faull, http://orcid.org/0000-0003-0897-7142

### Ethics

Human subjects: The Oxfordshire Clinical Research Ethics Committee approved the study and volunteers gave written, informed consent prior to participation.

## Additional files

### Supplementary files

• Supplementary file 1. Physiology and psychology of breathlessness task. (A) Supplementary Table 1. Mean (±sd) physiological variables across conditioned respiratory tasks. (B) Supplementary Table 2. Mean (±sd) physiological and psychological variables during breathlessness for both athletes and sedentary subjects. No significant differences were found between groups.

• Supplementary file 2. Data preprocessing and measures of residual motion. (A) Supplementary Table 3. Index of data quality and residual head motion using a framewise change in signal intensity. Mean and standard deviation (SD) of DVARS (root mean square of the differential of all timecourses within the mask at each frame) as a percentage of BOLD signal for raw and cleaned data (see below for details on data cleaning procedures). (B) Data preprocessing description.

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
