## [Decision Letter]

Thank you for submitting your article "The periaqueductal gray as the puppeteer in the human response to the threat of breathlessness?" for consideration by *eLife*. Your article has been reviewed by three peer reviewers, and the evaluation has been overseen by a Reviewing Editor and Eve Marder as the Senior Editor. The following individuals involved in review of your submission have agreed to reveal their identity: Hari Subramanian (Reviewer #1) and Nina De Lacy (Reviewer #3).

The reviewers have discussed the reviews with one another and the Reviewing Editor has drafted this decision to help you prepare a revised submission.

Summary:

This is potentially an important study, elucidating how PAG may orchestrate responses to respiratory threat. The study draws from its previous meritorious publication, and is thus a good candidate for an "Advance study". The authors perform a characterization of the activity and connectivity of individual PAG columns with higher cortical areas using 7T fMRI. The authors postulate that the connectivity with higher cortical areas represents a "breathlessness network" observable in the resting-state.

Subjects were specifically trained to perform a task inducing operant-conditioned emotional anticipation of breathlessness and free breathing conditions. Task-based fMRI is used to identify areas of the PAG that display time-correlated activity with anticipation of breathlessness. These areas are then used as seed regions to search for correlated activity with higher cortical areas during the resting state, averaging correlations across the resting-state time course.

The reviewers raised several major concerns with regards to the approach and the interpretation of the results, and arrived at a number of specific recommendations that need to be addressed by the authors.

Essential revisions:

1) Conceptually, it would be very helpful if the authors formulate specific and testable hypotheses in the Introduction. Hypotheses seem feasible and appropriate given that this is a ROI-based study with two seeds placed to construct activity maps, as opposed to a data-driven methodology. Indeed, the authors essentially report a two-stage study, where a task condition is used to generate hypothesized activation patterns that are subsequently analyzed in the resting state. This could be more clearly stated in the Introduction.

2) Along the same lines, the presentation of clear hypotheses will aid in the construction of robust links to the Discussion section and the interpretation of the results. In its current version, it is unclear whether the thrust of this paper is geared toward an analysis of 'causal connectivity' (the uncovering of causal or driving circuits) or 'functional connectivity' (statistical dependencies without the necessary implication of causality). The use of PPI as a core method suggests the former. However, the question of connectivity is a major issue of the paper and needs to be clearly defined.

Thus, we highly recommend that the authors present their study in a more focused manner, thereby framing their study as either a 'causal' study, or a 'functional' study, with clearly set forth hypotheses linking to methods, results and discussion. At its current state this led to considerable confusion that was also reflected in the other comments as stated below.

3) The authors propose that they have potentially uncovered a "breathlessness network" with two "pathways" associated with a) goal-directed movement and b) anxiety encoding that are specific to respiration. However, there is no evidence presented that these elements are not just consistent with canonical circuits involved in goal-directed activity and anxiety (see for example, the NIH RDoc framework related to "acute threat/fear" and "positive valence"). Thus, it is not clear that the postulated circuits are specific to respiratory control. Consequently, the authors need to soften the specificity of their claim, or perform additional experiments using other threat stimuli to delineate specificity to respiration. This is particularly relevant since the human data presented here are not entirely consistent with studies performed in other species using different, and potentially more precise approaches as will be also raised in a later comment (10).

4) Along the same lines: The experiments performed do not specifically test the role of PAG in breathing control. It also doesn't provide direct information from and what regions it projects to. Hence it is unclear what the hardware is that would justify calling the PAG "a puppeteer". The question of connectivity and the potential role of the PAG as the puppeteer will be further addressed by additional reviewers' comments (see below).

5) The authors should not forget (and they should therefore mention) that the PAG is not only involved in freezing, fight and flight, but also in controlling all other body activities as heart rate, blood pressure, vocalization, micturition, defecation etc. etc.

6) This study uses the psychophysical-interaction (PPI) method instantiated in FSL to draw conclusions about observed co-activation between the vlPAG and lPAG and cortical areas. PPI is a method aimed at uncovering causal connections, not just co-activation patterns, between brain regions, using linear regression analysis. A primary alternative method is dynamic causal modeling (DCM) a neuronal mass model founded in nonlinear dynamic methods. Many researchers follow Friston in considering DCM a superior method than either PPI or Grainger Causality since it addresses concerns such as directionality, the treatment of inputs as stochastic and the influence of the BOLD hemodynamic response transformation. Although the authors do not directly claim that they have identified roles for the vlPAG and lPAG in causally driving a neuronal system associated with respiratory control, they speak of "connections". In the Discussion section the authors mention the possibility of the vlPAG in an "orchestra(l)" role and the title clearly implies causality in proposing the PAG as "puppeteer". They clearly need to soften their claims. Indeed, it is not entirely clear why the authors chose to use PPI – a causal method – and did not just explore what has been called "functional connectivity" or the statistical dependencies between regions that may or may not be causal. If the authors had an interest in examining causality but did not find it, they should report this and as well, explain their choice of PPI over DCM. If they were instead primarily interested in exploring functional connections, they should perhaps consider a simpler and less controversial method. If they believe their results do, in fact, demonstrate causality, then they should clearly present this and again make the case for PPI over DCM.

7) Along the same line: In the title the authors think of the PAG as the puppeteer of the response to breathlessness. As discussed in the previous paragraph, the experimental design and approach does not really test this role. The authors need to clearly soften their language and change the title and Discussion accordingly.

8) The authors present an analysis of connections observed in resting-state fMRI. While similar patterns of co-activation often deemed "networks" may be seen in the task and resting states, it is not yet quite clear what circuits observed during rest in fact represent. The authors should contextualize this. Moreover, it is unclear what the role of the resting-state analysis is in this study. The authors need to clearly state their purpose in providing resting-state results. The authors claim particular "networks" are identified by the placement of two seeds and examining patterns of co-activation with these seeds across the cortex in the averaged timeseries. However, it is not clear that seed-based methods reliably identify stable or replicable "networks" at rest without bias, and this concern is particularly heightened when the researchers are proposing the discovery of a hitherto undiscovered, novel "network".

Indeed, one can get fairly comfortable with seed-based methods if a widely confirmed, major network such as the dorsal attention or default mode network is the unit of analysis where specific seeds such as the frontal eye fields or posterior cingulate (respectively) are well-accepted. However, in the present study the authors propose discovering a novel network by manual seed placement. Generally, a data-driven or hypothesis-free method would be a preferred approach to accomplish this aim since this is a more robust and bias free method (see for example methodologic comparisons by Raichle and others).

9) The brain and one presumes respiration, is inherently dynamic. Thus, the authors might comment in their Discussion on why averaged co-activation across the timecourse (aka "static connectivity") is adequate to describe a 'respiratory network' in the resting state or what the scope for further investigation or work might be around these concepts arising from their investigations. Certainly, this discussion would be quite interesting for a general readership.

10) Along the same lines, defining the conditions of the experimental paradigm is crucial. What the authors mean with "at rest" is really not at "rest" but rather "anticipation of breathlessness". Please clarify in the text.

11) fMRI is in general quite sensitive to head motion and head motion can easily give spurious effects in both task and rest conditions. This has become increasingly a concern for example in developmental studies in which children may have produced more movements that more adult subjects. The authors mention that "motion correction and motion parameter recording" was performed. However, the reviewers were unable to find in the Methods section that motion correction information for subjects was modeled as a regressor in any level of the statistical analysis. It would also be important to know as to what residual motion remained after correction and how this was incorporated in the statistics. Some helpful methods for this (see for example Powers et al.) have been suggested. The reviewers suggest performing the analysis after incorporating specific motion correction parameters as regressors as well as analyzing the data for residual motion.

12) The authors consistently use the term "connectivity" without providing a clear definition in this paper. The authors will agree that this term can have substantially different connotations in different contexts and moreover be imperfectly understood by scientists and readers outside the fMRI field. The reviewers urge the authors to use more precise terms, or defining a precise meaning for "connectivity" within this current study.

13) Along the same lines. In Figure 3 the authors show vlPAG and lPAG resting connectivity. The authors need to specify what exactly do they compare? In the same figure the authors show the PAG as occupying almost half of the midbrain, which cannot be the case: please address this issue. Also, in the Abstract, the authors state that the lPAG was functionally connected with sensorimotor areas (presumably the cortex). What do the authors specifically mean with connected? It is unlikely that the authors want to imply that the PAG has any direct afferent of efferent connections with the sensorimotor cortical areas. One has to assume that another region in the brain might influence these areas, which does not mean that the PAG is connected with the sensorimotor cortical areas. Thus, as already stated in the prior comment, the authors are urged to clearly specify what they mean with "connectivity".

14) The present study does not show or mention the dorsal PAG. Since this study deals with the interaction between individual PAG columns and cortical structures in the context of respiratory threat, can the authors discuss why they do not see any activation or deactivation of the dorsal PAG in this overall scheme? The dorsal PAG has been suggested to play a strong role in subsequent sensory processing during and following a somatic/autonomic outflow. The dorsal PAG has also been implicated in pain processing, afferent visceral representation and defense reactions. Phantom limb pain human patients often experience dyspnea. Recent animal studies have shown subpopulation of neurons in the dorsal PAG responding freezing and avoidance. A recent study by Satpute et al. showed dorsomedial PAG activation to threat and stress in humans.

15) Main text, second paragraph: What are the premotor areas in the ponto-medullary reticular formation? How is this circuitry built? See the comments regarding connectivity above.

Maybe it might be helpful for the reader to add a summary diagram, showing the incoming and out-going pathways of the different parts of the PAG. The present version of this paper generates a lot of question marks.

16) There are species-specific differences in the functional topography, which the authors need to consider in interpreting the results from humans, using imaging. Topographic chemical or optical stimulation have led to detailed insights in columnar specificity that is not entirely consistent with this human study. Both Bandler, & Carrive labs (Carrive, 1993, an important citation missing)/Subramanian and Holstege labs have shown the lateral PAG to be involved in tachypnea. In the cat this leads to flight and as per Bandler/Carrive lab findings, also fight. While in the rat, the lPAG induced tachypnea may well correspond to dyspnea or fright. Freezing however is more caudal and ventrolateral (Subramanian & Holstege 2013a/b, 2014) characterised by distinct apnea/ breath-hold. One could assume that the increase in anxiety levels could affect fright towards freezing per se, thus a breath-hold or apnea, which would then increase the activity in the vlPAG rather than lPAG. However, the results reported by the authors are contrary. Could the authors comment on this? In particular, in light of the experimental approach that provides only limited insights into the causal relationship to respiration as raised in one of the first comments.

17) Anticipation & breathlessness: Was any activity seen in the bed nucleus? Particularly when the central nucleus of the amygdala together with the lateral bed nucleus of the stria terminalis projecting strongly to the PAG, one would assume a strong activation of the bed strialis. Was this seen?

18) The authors report on the resting functional connectivity of the vlPAG to the ACC. Did the authors see any PAG columnar interactions with the ACC during either breathlessness or its anticipation?

19) Subsection “lPAG in the fight/flight response to threat” seems a bit far-fetched to me, particularly the inference of 'plausible pathways' for goal-directed behaviours. Does an 'escape' response either via flight or fright involve 'goal directed behaviour"? It would seem more an innate response to me, although the functional connectivity with the caudate is of significant interest no doubt.

20) Holstege & Hopkins showed in the cat that wiring includes amygdala-bed nucleus triangulation into the PAG. It's a bit of a long shot to suggest that anxiety-encoding involves just the amygdala-PAG axis in humans (that to cross-referenced to work on transgenic mice) despite the pronouncement of the terminalis area.

21) Anticipatory respiratory threats would involve learning mechanisms which brings in the hippocampal-amygdaloid interaction. Was any lighting up seen in any of the hippocampal regions?

22) While the Tovote et al. optogenetic study provides salient details of the amygdala-PAG interaction, their identification of a group of cells what they call "magnocellular nucleus of the medulla" seems strange. The area they identify in the medulla points more or less to ventrolateral medullary cells, a region involved in cardiorespiratory control, rather than locomotor control.

23) Studies by the Subramanian lab demonstrated direct modulation of the respiratory cells by the PAG in inducing various respiratory effects. Schimitel and Gargaglioni labs also suggested hypoxia-induced reduction and avoidance underplays freezing behavior. They also involved the VLM cardiorespiratory cells. In another important study, Oka et al. (2008) demonstrated that GABA-ergic neurons in the central nucleus of the amygdala possibly influence the PAG-NRA pathway in mediating fear. Thus, the authors need to carefully pitch their reasoning for examining the ponto-medullary structures.

24) Subsection “Intra-PAG connectivity and interaction between fight/flight and freeze responses”, combines species across spectrum from the rat (Jansen et al.) to the transgenic mice (Tovote). The columnar structure of the PAG as defined in cat and to a certain extent in rat has been simply incorporated to the transgenic mice. The rostrocaudal extension of the PAG defines at large the columnar structure (Carrive) identified mainly via behavioural and autonomic topography. In the mice, this fundamental information is forthcoming. The discussion on this section is largely based upon reciprocal inhibition suggested based on anatomical wiring (Jansen) and the study by Tovote. Lack of concrete human data, means this can be simplified rather than relying heavily on cross-species animal work.

25-28) The reviewers also had various methodological comments that need to be addressed:

25) The authors make many specific methodologic choices in their pre-processing. Some of these are less common (for example, a 2mm Gaussian kernel) and some have their believers and non-believers (for example, field unwarping). The reviewers appreciate the reference to a previous recent paper by the same first author, but the authors should also comment (briefly) on some of the more prominent choices. As well, if there are any specific choices made on the basis of imaging at 7T strength it would be helpful to contextualize this.

26) The authors here perform essentially a cluster based task-fMRI analysis. Recently, these methods have come under substantial scrutiny in the wider neuroscience community due to their vulnerability to serial autocorrelation. The reviewers suggest that the authors acknowledge this, and explain in detail how their use of correction in FILM, as well as the significance thresholds they employ in their statistical analysis, adequately address these concerns. For example, the reviewers noted that the authors use a significance threshold of p<0.05 to perform cluster testing.

Please define correction methods used during significance testing.

27) There are many FSL-specific abbreviations used in the Methods section. They have the cumulative effect of diminishing the overall clarity and understanding. The reviewers suggest defining that these are toolboxes within FSL and specifying what they are before acronyms are used. As well, a little too much is assumed in terms of how various toolboxes in FSL perform these various components of processing and analysis. In our community, various choices of software exist, and each involves some embedded methodologic choices. It is helpful to at least briefly but clearly state what is implied by the choice of each major step in the FSL pipeline.

28) The reviewers encourage the authors to provide a spreadsheet as a supplement including, on a per subject basis, the regressors used as well as measures of head motion, with brief summary statistics (e.g. mean and SD).

[Editors' note: further revisions were requested prior to acceptance, as described below.]

Thank you for submitting your article "The cortical connectivity of the periaqueductal gray and the conditioned response to the threat of breathlessness" for consideration by *eLife*. Your article has been reviewed by two peer reviewers, and the evaluation has been overseen by a Reviewing Editor and Eve Marder as the Senior Editor. The following individuals involved in review of your submission have agreed to reveal their identity: Hari Subramanian (Reviewer #1) and Nina De Lacy (Reviewer #3).

The reviewers have discussed the reviews with one another and the Reviewing Editor has drafted this decision to help you prepare a revised submission.

Summary:

This is a revised manuscript of an advance article in which the authors characterized the activity and connectivity of individual PAG columns with higher cortical areas using 7T fMRI. The goal of the study is to identify under resting state conditions a breathlessness network that connects the PAG with higher cortical areas.

The authors have done a nice job of addressing concerns regarding the presentation, phrasing and computation of their results with respect to the vast majority of the neuroimaging findings. However, the reviewers had some remaining comments that need to be addressed.

Essential revisions:

1) One set of comments relates to the handling of motion correction. The authors have regressed out variance they identified with motion. It will be important to specify whether this was done at the first or second level analysis.

However, the bigger issue is whether this study can exclude residual motion after denoising? This can be accomplished, for example, by including motion regressors in the statistical analysis at the second level. Since the authors have already calculated their DVARs measures this should be relatively fast. They would then have the opportunity to declare their significant results were unaffected by residual motion.

2) In response to comment 14, the authors state: "The Satpute paper does mention dmPAG activity in the text of its results, the main figure and data presented in that paper does not support this; instead demonstrating main activity in the vlPAG and bordering lPAG whilst subjects viewed threatening images, with no evidence for dmPAG activity. […] We have now included a section within the limitations section of the discussion to address the dorsal PAG (or lack thereof) in our study, with the following text".

However, the text (in subsection “Limitations”, first two paragraphs) added to the manuscript does not really address the issue of dorsal PAG (or lack thereof)! The authors have merely stated a well-known limitation of the fMRI method. Indeed, fMRI method does not provide topographic specificity of the columns, or the effect seen can be interpreted as causal to the stimuli. Stating this does not answer the question in the context of the dorsal PAG.

The authors should discuss in the Discussion section the reasoning in terms of the structure of the human PAG (cylindrical vs. any other published /assumed), the lack of border differentiation of the human PAG, and why this was seen, noted or inferred on in other studies (say Satpute et al) and not in the present study. The current results are similar to the earlier *eLife* paper and would just remain an incremental research advance if avenues such as the lack of activation of the dorsal PAG, a major center implicated in subsequent sensory processing during and following a somatic/autonomic outflow is not discussed adequately.

3) Related to comment 16: We agree that "studying humans gives us the unique opportunity to ask individuals their subjective ratings of emotions such as anxiety, rather than inferring these from physiological and behavioural measures in animals". Irrespective of whether one examines by direct stimulation of PAG columns or assessing signal activation of the columns indirectly, the authors are dealing with a set of highly specialized circuitry, topographically organized for appropriating specific physiological and behavioural responses. Given the non-specificity of the fMRI method vs. highly specific animal methods, I would consider it more important to discuss the findings in humans against animal observations (with appropriate citations, as key references are missing) for inference on PAG function regarding respiratory threat in humans. We suggest that the authors add a separate paragraph to examine human findings against animal studies that have examined the physiology and behavioural responses to respiratory threat.

4) Related to the above and comment 23. The various (animal) studies that we listed deal with breathlessness that can be induced from the PAG, its connotations to suffocation alarm, fear and perception of respiratory threat. Since the current study deals with the threat of breathlessness (and not a generic fear mediation circuitry as investigated by Tovote et al), a discussion linking the human findings to animal studies in the context of respiratory threat is imperative. Particularly when the authors infer upon the inhibition of lPAG and its downstream connections inhibiting the flight/fight to instigate freezing which are clearly elucidated in animal models.

---

## [Author Response]

*Summary:*

*This is potentially an important study, elucidating how PAG may orchestrate responses to respiratory threat. The study draws from its previous meritorious publication, and is thus a good candidate for an "Advance study". The authors perform a characterization of the activity and connectivity of individual PAG columns with higher cortical areas using 7T fMRI. The authors postulate that the connectivity with higher cortical areas represents a "breathlessness network" observable in the resting-state.*

*Subjects were specifically trained to perform a task inducing operant-conditioned emotional anticipation of breathlessness and free breathing conditions. Task-based fMRI is used to identify areas of the PAG that display time-correlated activity with anticipation of breathlessness. These areas are then used as seed regions to search for correlated activity with higher cortical areas during the resting state, averaging correlations across the resting-state time course.*

*The reviewers raised several major concerns with regards to the approach and the interpretation of the results, and arrived at a number of specific recommendations that need to be addressed by the authors.*

We firstly thank you for your interest in our work, and for the thorough and very helpful review. We have attempted to address all concerns to improve clarity and validity of our results. We would firstly like to clarify that while we have indeed used task-based fMRI to identify areas of activity in the PAG (and cortex) during both anticipation of breathlessness and during breathlessness itself, we have then gone on to test functional connectivity of these PAG seeds in two settings; at rest and during the anticipation and breathlessness tasks.

We have employed seed-based functional connectivity (using correlated BOLD activity) between these seeds and the wider cortex during resting state. The PPI approach used measures correlated activity, but does not infer causality. For our task-based functional connectivity method, we have used a psychophysiological interaction (PPI) analysis, which is a similar method of correlated ‘functional connectivity’ between brain areas, but specific to each task condition and without the temporal filtering complications that come when splitting functional data into blocks and running separate correlations for each task.

We would like to clarify, and apologise for not making it clear that we have run a functional connectivity analysis in both cases, rather than a causal (or effective) connectivity analysis. We have now made it clearer in the manuscript that directionality cannot be ascertained with our methods, so to avoid potential misunderstanding. We have added more detail to this effect in the Introduction and Methods etc., and hope these revisions address the reviewers’ questions.

*Essential revisions:*

*1) Conceptually, it would be very helpful if the authors formulate specific and testable hypotheses in the Introduction. Hypotheses seem feasible and appropriate given that this is a ROI-based study with two seeds placed to construct activity maps, as opposed to a data-driven methodology. Indeed, the authors essentially report a two-stage study, where a task condition is used to generate hypothesized activation patterns that are subsequently analyzed in the resting state. This could be more clearly stated in the Introduction.*

Many thanks for this comment. We agree that this would be a helpful addition to the Introduction. With the fluidity of the guidelines on the Research Advance we had kept this predominantly in our original submission, but now see that this is disadvantageous for the reader of this manuscript. We have now added the following text to the Introduction:

“In this study we wanted to further test the hypothesis that the vlPAG is involved with learned anticipatory threat detection of breathlessness, whilst the lPAG is associated with the active response to breathlessness itself, by investigating how these PAG regions interact with the wider cortex. […] Measures of this connectivity both at rest and during a specific task can be used to investigate the dynamic, task-specific changes in functional connectivity between regions […]”

*2) Along the same lines, the presentation of clear hypotheses will aid in the construction of robust links to the Discussion section and the interpretation of the results. In its current version, it is unclear whether the thrust of this paper is geared toward an analysis of 'causal connectivity' (the uncovering of causal or driving circuits) or 'functional connectivity' (statistical dependencies without the necessary implication of causality). The use of PPI as a core method suggests the former. However, the question of connectivity is a major issue of the paper and needs to be clearly defined.*

*Thus, we highly recommend that the authors present their study in a more focused manner, thereby framing their study as either a 'causal' study, or a 'functional' study, with clearly set forth hypotheses linking to methods, results and discussion. At its current state this led to considerable confusion that was also reflected in the other comments as stated below.*

Thank you for identifying this important point of confusion in our manuscript. As explained, we have used a functional connectivity approach (now outlined in the Introduction of the manuscript) that neither tests nor infers causality.

We agree with the reviewers that measures of functional connectivity might be the best approach to this early study of PAG connectivity in humans, as has recently become possible with high-resolution, high-power 7 Tesla scanning. Our resting state data was analysed using a region of interest (ROI) driven approach of the active PAG seeds, whilst the task-based functional connectivity was conducted using psychophysiological interaction (PPI) modelling.

PPI is similar to a resting state ROI timecourse-driven correlation analysis, but simply uses an interaction term to test whether the ‘functional connectivity’ or correlation is stronger in a specific task to that of the other tasks in the study. This avoids the need to isolate the set of timecourses associated with each task, which negatively affects both temporal smoothing and temporal filtering applied to fMRI data, and also creates ambiguity when task activity is not time-locked to the beginning of a volume acquisition (TR) (as occurred in this case, as ‘jittering’ between the TR and repeated task blocks is encouraged to allow for accurate sampling across the entire haemodynamic response). PPI also allows us to distinguish a change in ‘functional connectivity’ between remote brain areas from a simple concurrent increase in activity between these areas. The following text has also been added to the Methods section to more clearly explain this technique to non-imaging experts, and we hope this helps to clarify our methods.

“Task functional connectivity analyses: Functional connectivity between the PAG and the rest of the brain during the task conditions of anticipation and breathlessness was measured using a psychophysiological interaction (PPI) analysis (Friston 1997, OReilly 2012) for both of the task-active PAG columns. […] A pre-threshold grey-matter mask was applied to constrain our analysis to connectivity of the PAG seeds to the cortical and subcortical grey matter.”

*3) The authors propose that they have potentially uncovered a "breathlessness network" with two "pathways" associated with a) goal-directed movement and b) anxiety encoding that are specific to respiration. However, there is no evidence presented that these elements are not just consistent with canonical circuits involved in goal-directed activity and anxiety (see for example, the NIH RDoc framework related to "acute threat/fear" and "positive valence"). Thus, it is not clear that the postulated circuits are specific to respiratory control. Consequently, the authors need to soften the specificity of their claim, or perform additional experiments using other threat stimuli to delineate specificity to respiration. This is particularly relevant since the human data presented here are not entirely consistent with studies performed in other species using different, and potentially more precise approaches as will be also raised in a later comment (10).*

Thank you for this comment – we can now see that the way we have explained this may be cause confusion. We had not intended to claim that the functionally connected regions that we labelled as a ‘network’ were either 1) a novel ‘network’ in the framework of previously-identified resting state ‘networks’; nor 2) specific to respiratory control. We instead considered the term ‘network’ to indicate simply a set of functionally connected regions, but can see that the connotations with previous resting state literature may imply this for some readers.

We have now changed our use of the term ‘network’ to a ‘system’ or a ‘functionally connected set of regions’ (or variations of) throughout the text, and hope that this is now much clearer for readers. Therefore, we hope this will also soften our (unintended) claim that we have discovered a novel ‘network’ of breathlessness.

We referred specifically to the threat of breathlessness in this paper as we looked at both activity and functional connectivity of the PAG using a threatening breathlessness task, and did not want to over-generalise without testing other types of threat. We agree that this could produce similar findings in other types of threat, and look forward to future research in other areas to investigate this. We have now softened our claims in relation to the term ‘breathlessness / respiratory threat’, and have added the following comment to the Discussion:

“We do not yet know whether these findings are specific to respiratory threat or generalizable. Therefore, future work investigating functional connectivity of the substructures of the PAG could employ alternative threatening stimuli, for example pain, to extend our understanding of the role of the PAG columns across generalized threat responses.”

*4) Along the same lines: The experiments performed do not specifically test the role of PAG in breathing control. It also doesn't provide direct information from and what regions it projects to. Hence it is unclear what the hardware is that would justify calling the PAG "a puppeteer". The question of connectivity and the potential role of the PAG as the puppeteer will be further addressed by additional reviewers' comments (see below).*

We agree that using the word ‘puppeteer’ does muddy the waters of a functional connectivity study, and potentially imply causality when this was never tested in this analysis. We have now changed the title of the paper and removed references to a ‘puppeteer’ in the manuscript.

*5) The authors should not forget (and they should therefore mention) that the PAG is not only involved in freezing, fight and flight, but also in controlling all other body activities as heart rate, blood pressure, vocalization, micturition, defecation etc. etc.*

Thank you for this comment. We agree that relying too heavily on the information presented in the paper prior to a Research Advance might narrow the focus of the paper too much, and remove important context when an Advance is read in isolation. We have now prefaced our Introduction to the hypothesised role of the PAG in threat behaviours with the following text:

“While the PAG has previously been implicated in many basic survival functions including cardiovascular, motor and pain responses such as vocalization or blood pressure regulation (DeOca 1998, Mobbs 2007, Pereira 2010, Tracey 2002, Paterson 2013), an integrative theory of these behaviours postulates that the lPAG and dlPAG are thought to orchestrate ‘active’ responses (such as fight or flight responses) when a threat is perceived as escapable (Bandler 1988, Depaulis 1992, Yardley 1986, Keay 2001). Conversely, the vlPAG is thought to be involved with ‘freezing’ type behaviours from inescapable threats (Tovote 2016, Lovick 1993, Keay 1997, Carrive 1991), including conditioned anticipation of breathlessness in humans (Faull 2016).”

*6) This study uses the psychophysical-interaction (PPI) method instantiated in FSL to draw conclusions about observed co-activation between the vlPAG and lPAG and cortical areas. PPI is a method aimed at uncovering causal connections, not just co-activation patterns, between brain regions, using linear regression analysis. A primary alternative method is dynamic causal modeling (DCM) a neuronal mass model founded in nonlinear dynamic methods. Many researchers follow Friston in considering DCM a superior method than either PPI or Grainger Causality since it addresses concerns such as directionality, the treatment of inputs as stochastic and the influence of the BOLD hemodynamic response transformation. Although the authors do not directly claim that they have identified roles for the vlPAG and lPAG in causally driving a neuronal system associated with respiratory control, they speak of "connections". In the Discussion section the authors mention the possibility of the vlPAG in an "orchestra(l)" role and the title clearly implies causality in proposing the PAG as "puppeteer". They clearly need to soften their claims. Indeed, it is not entirely clear why the authors chose to use PPI – a causal method – and did not just explore what has been called "functional connectivity" or the statistical dependencies between regions that may or may not be causal. If the authors had an interest in examining causality but did not find it, they should report this and as well, explain their choice of PPI over DCM. If they were instead primarily interested in exploring functional connections, they should perhaps consider a simpler and less controversial method. If they believe their results do, in fact, demonstrate causality, then they should clearly present this and again make the case for PPI over DCM.*

Thank you for this important discussion point. As explained, we have not tested causality using an effective connectivity method, and hope that the revised manuscript explains this more clearly. Within the PPI methods used in this manuscript, an ROI of each PAG region was used to generate a timeseries plot, and the correlations with this timeseries were assessed across each grey matter voxel of the brain.

We agree that methods used to infer causality with fMRI data, such as Grainger causality, are rife with assumptions and are hotly debated. PPI is also less subjected to issues regarding BOLD haemodynamic transformation, as the timeseries for the physiological regressor is taken from the BOLD data itself, and thus there is no need for a (de)convolution of any kind.

We also agree with the reviewers that a measure of ‘functional connectivity’ is a more conservative and appropriate approach to this study, rather than using a dynamic causal modeling approach for causality, and would like to clarify that this was never attempted with this data. We hope that our further explanations of the methods used now address this issue, allowing our manuscript to be better understood.

We have also made sure to remove or soften any claims that could be misinterpreted towards causality, such as notions of a ‘puppeteer’ or the PAG columns playing an orchestral role in behavioural responses.

*7) Along the same line: In the title the authors think of the PAG as the puppeteer of the response to breathlessness. As discussed in the previous paragraph, the experimental design and approach does not really test this role. The authors need to clearly soften their language and change the title and Discussion accordingly.*

We agree that the notion of the PAG as a puppeteer cannot be claimed from this data, and have changed the title of the manuscript accordingly.

*8) The authors present an analysis of connections observed in resting-state fMRI. While similar patterns of co-activation often deemed "networks" may be seen in the task and resting states, it is not yet quite clear what circuits observed during rest in fact represent. The authors should contextualize this. Moreover, it is unclear what the role of the resting-state analysis is in this study. The authors need to clearly state their purpose in providing resting-state results. The authors claim particular "networks" are identified by the placement of two seeds and examining patterns of co-activation with these seeds across the cortex in the averaged timeseries. However, it is not clear that seed-based methods reliably identify stable or replicable "networks" at rest without bias, and this concern is particularly heightened when the researchers are proposing the discovery of a hitherto undiscovered, novel "network".*

Thank you for this comment. We investigated the functional connectivity of these PAG seeds both at rest and during breathlessness tasks to explore the dynamic nature of this connectivity, as resting state connections may not translate to more specific task-based functional connectivity. We have now added the following text to the Introduction to address this point:

“Functional connectivity’ within neuroimaging is a measure of the temporal synchronicity of activity within structures across the brain, and is thought to be related to the temporal coherence of neuronal activity in anatomically distinct regions (Gerstein and Perkel, 1969; Van Den Heuvel and Pol, 2010). […] The findings of this study will help us to understand how the different roles of the individual PAG columns interact with the wider cortex in humans, both at rest and when perceiving a threatening stimulus such as breathlessness.”

*Indeed, one can get fairly comfortable with seed-based methods if a widely confirmed, major network such as the dorsal attention or default mode network is the unit of analysis where specific seeds such as the frontal eye fields or posterior cingulate (respectively) are well-accepted. However, in the present study the authors propose discovering a novel network by manual seed placement. Generally, a data-driven or hypothesis-free method would be a preferred approach to accomplish this aim since this is a more robust and bias free method (see for example methodologic comparisons by Raichle and others).*

We apologise for any confusion caused by our use of the term ‘network’, and can now see how this could be interpreted against the backdrop of previous resting state analyses where this term has gained in momentum. As mentioned, we were not trying to infer the discovery of a novel network of breathlessness within either our resting state or task analysis – as rightly pointed out, a data-driven approach would be the robust way of doing this, although at rest there would be no specific tie to breathlessness and thus would be very speculative indeed.

What we were attempting to achieve was a seed-based, functional connectivity analysis of the active PAG areas both at rest, and during the tasks of both breathlessness anticipation and the experience of breathlessness itself. We agree that these PAG functional connectivity profiles do not reveal a novel network, nor can this group of functionally connected regions (especially at rest) be tied specifically to respiratory control, and have adjusted the manuscript to clarify this. The purpose of including a resting state analysis was to compare this set of functionally connected regions to how the connectivity profile changes when in a task state. We have now adjusted our main findings section of the Discussion to the following text:

“In this study we have demonstrated vlPAG and lPAG activity during conditioned anticipation of breathlessness, and lPAG activity during breathlessness that scaled with subjective scores of breathlessness intensity. […] Furthermore, during anticipation of breathlessness, the vlPAG demonstrated reduced connectivity to both the lPAG and sensorimotor structures, consistent with a potential ‘freeze’ response.”

*9) The brain and one presumes respiration, is inherently dynamic. Thus, the authors might comment in their Discussion on why averaged co-activation across the timecourse (aka "static connectivity") is adequate to describe a 'respiratory network' in the resting state or what the scope for further investigation or work might be around these concepts arising from their investigations. Certainly, this discussion would be quite interesting for a general readership.*

Thank you for this interesting point. Dynamic functional connectivity that tightly follows changes in the dynamic respiratory response would be of great interest when investigating both generalised threat and responses to a respiratory threat such as breathlessness.

However, our research question was to investigate both the activity and connectivity of the PAG and cortex during the threat of breathlessness, which is not time-locked to the respiratory cycle. Whilst we were also not aiming to describe a ‘respiratory network’ in the resting state, as we hope to have now clarified, connectivity changes directly related to changes in respiration would be of interest in future research. In the present experiment, we prioritised resolution and whole-brain coverage at the sake of a longer TR (repetition time, or time to measure one brain volume), which limits our temporal resolution and the ability to determine brain activity related to any short-period cycles, such as respiration. Additionally, our noise correction techniques were prioritised to minimise physiological noise in the PAG, and these techniques remove signal that cannot be teased apart from the respiratory cycle. Thus, as it currently stands, dynamic connectivity that parallels respiratory control is beyond the current experiment. We have now addressed this in a limitations section in the Discussion to raise these points:

“The temporal blurring of BOLD signal due to the HRF, coupled with the long (3 second) volume repetition time (TR) employed in this study would mean vastly greater statistical power would be likely necessary to attempt either intricate causality modelling (‘effective connectivity’) using methods such as dynamic causal modelling (DCM) (Friston et al., 2003), or more dynamic versions of functional connectivity (Chang and Glover, 2010). Whilst techniques such as simultaneous multi-slice imaging are now becoming more mainstream (Feinberg et al., 2010; Moeller et al., 2010), researchers may need to be wary of potential blurring between slices deep in the brain within image reconstruction of these acquisitions (Feinberg et al., 2010), which may temper our ability to functionally isolate structures such as the subdivisions of the PAG.”

*10) Along the same lines, defining the conditions of the experimental paradigm is crucial. What the authors mean with "at rest" is really not at "rest" but rather "anticipation of breathlessness". Please clarify in the text.*

We have now added the following text to the brief overview that prefaces the Results section:

“Forty healthy, right-handed individuals were trained using an aversive delay-conditioning paradigm to associate simple shapes with either a breathlessness stimulus in the form of inspiratory resistive loading (100% contingency pairing) or no loading (0% contingency pairing). […] Task fMRI data was analysed for mean changes in BOLD activity, and functional activity identified in both the vlPAG and lPAG was then subsequently analysed for functional connectivity with the wider cortex at rest (using the resting state scan), and during anticipation and breathlessness (using the task fMRI scan).”

*11) fMRI is in general quite sensitive to head motion and head motion can easily give spurious effects in both task and rest conditions. This has become increasingly a concern for example in developmental studies in which children may have produced more movements that more adult subjects. The authors mention that "motion correction and motion parameter recording" was performed. However, the reviewers were unable to find in the Methods section that motion correction information for subjects was modeled as a regressor in any level of the statistical analysis. It would also be important to know as to what residual motion remained after correction and how this was incorporated in the statistics. Some helpful methods for this (see for example Powers et al.) have been suggested. The reviewers suggest performing the analysis after incorporating specific motion correction parameters as regressors as well as analyzing the data for residual motion.*

We can confirm that motion regressors were incorporated into the analysis. These regressors were included in the denoising preprocessing procedures that encompassed both ICA denoising and RETROICOR. Motion regressors calculated from the motion correction preprocessing step (using MCFLIRT in this case) were regressed out of the data alongside the ‘noise’ components identified during ICA denoising. We agree that adequate motion correction is fundamentally important, (particularly) when attempting any functional connectivity analyses, as spurious correlations could greatly affect results. We have now added this text to the Methods section to explain this procedure:

“Due to the location of the PAG and its susceptibility to physiological noise (Brooks et al., 2013), rigorous data denoising was conducted using a combination of independent components analysis (ICA) and retrospective image correction (RETROICOR) (Harvey 2008, Brooks 2013), as previously described (Faull 2016). […] Physiological recordings of heart rate and respiration (from respiratory bellows) were transformed into regressors (3 cardiac, 4 respiratory harmonics, and an interaction term (Harvey 2008)) corresponding to each acquisition slice, and the signal associated with this noise was isolated using linear regression, adjusted for any interaction with previously-identified ICA noise components and then subtracted from the data.”

We have also calculated a measure of residual motion for each subject to demonstrate the reduction in noise resulting from our preprocessing techniques, and have presented the results in a supplemental table. There are two common ways to measure residual motion – one is to sum the motion correction parameters, while the other is to calculate intensity differences between realigned volumes. The first is typically done using measures such as framewise displacement (an average of rotation and translation parameter differences calculated from motion correction), while the latter can be calculated using measures such as DVARS (D referring to temporal derivative of timecourses, VARS referring to Root Mean Square (RMS) variance over voxels).

Measures such as framewise displacement are dependent on the accuracy of the initial motion correction, and thus may not fully describe the residual motion. Alternatively, a DVARS measure roughly equates to the percentage signal change between volumes, and while this will also incorporate signal changes associated with tasks, it is thought to be more conservative as it is not reliant on the accuracy of the motion correction. We have calculated the mean and standard deviation of the DVARS for our unprocessed data and our final (ICA + RETROICOR) cleaned data, and presented this in Supplementary file 3. Powers et al. (2012) use a threshold of 0.5% BOLD signal change as an acceptable value for DVARS measurements, and our data shows each subject’s motion as below this threshold following rigorous preprocessing. This has now been explained in the Supplementary Material.

*12) The authors consistently use the term "connectivity" without providing a clear definition in this paper. The authors will agree that this term can have substantially different connotations in different contexts and moreover be imperfectly understood by scientists and readers outside the fMRI field. The reviewers urge the authors to use more precise terms, or defining a precise meaning for "connectivity" within this current study.*

A clear definition has now been added the following text to the Introduction:

“Functional connectivity’ within neuroimaging is a measure of the temporal synchronicity of activity within structures across the brain, and is thought to be related to the temporal coherence of neuronal activity in anatomically distinct regions (Gerstein and Perkel, 1969; Van Den Heuvel and Pol, 2010). […] Measures of this connectivity both at rest and during a specific task can be used to investigate the dynamic, task-specific changes in functional connectivity between regions, but do not infer directionality or causality.”

*13) Along the same lines. In Figure 3 the authors show vlPAG and lPAG resting connectivity. The authors need to specify what exactly do they compare? In the same figure the authors show the PAG as occupying almost half of the midbrain, which cannot be the case: please address this issue.*

The PAG in this figure (and all others) is a 3-dimensional projection of the whole PAG nucleus, while the rest of the midbrain has been left as 2-dimensional cut-outs. The lighter section of the PAG at the bottom shows the 3-dimensional PAG intersection with the 2-dimensional slice in the lower section of the midbrain. We have now added this notation to our figure legend (Figure 3), and explained the resting connectivity measures used here in addition to the Methods section.

If the reviewers feel that the PAG is still not clear as a 3-dimensional projection we are happy to alter this to a 2-dimensional image, however we hoped to provide a clearer demonstration of the seed placement within the PAG according to both its columnar and rostro-caudal position by presenting it within a 3-dimensional image.

*Also, in the Abstract, the authors state that the lPAG was functionally connected with sensorimotor areas (presumably the cortex). What do the authors specifically mean with connected? It is unlikely that the authors want to imply that the PAG has any direct afferent of efferent connections with the sensorimotor cortical areas. One has to assume that another region in the brain might influence these areas, which does not mean that the PAG is connected with the sensorimotor cortical areas. Thus, as already stated in the prior comment, the authors are urged to clearly specify what they mean with "connectivity".*

We hope that our clarifications regarding the term ‘functional connectivity’ will help to address this issue. Indeed, we were not insinuating any direct afferent or efferent connections between the PAG and sensorimotor areas of the cortex, as this could most definitely not be determined using our method, or any others outside of direct tracer methodologies. We have now adjusted this statement in the Abstract to the following:

“At rest the lPAG was functionally correlated/connected with cortical sensorimotor areas”

*14) The present study does not show or mention the dorsal PAG. Since this study deals with the interaction between individual PAG columns and cortical structures in the context of respiratory threat, can the authors discuss why they do not see any activation or deactivation of the dorsal PAG in this overall scheme? The dorsal PAG has been suggested to play a strong role in subsequent sensory processing during and following a somatic/autonomic outflow. The dorsal PAG has also been implicated in pain processing, afferent visceral representation and defense reactions. Phantom limb pain human patients often experience dyspnea. Recent animal studies have shown subpopulation of neurons in the dorsal PAG responding freezing and avoidance. A recent study by Satpute et al. showed dorsomedial PAG activation to threat and stress in humans.*

The reviewers raise a very interesting and relevant point. We agree that there is important literature identifying the role of the dorsal PAG in sensory processing and defence reactions. Although we did not observe dorsal PAG activity in our study, we did, however, observe changes in connectivity between lPAG and dorsal PAG (Figure 4) that inversely correlate with behavioural scores of breathlessness intensity. We have highlighted this finding within our Discussion (see text below). However, without robust changes in BOLD activity of the dorsal PAG as well as changes in connectivity, we feel it is appropriate to be cautious in its interpretation at this point, and therefore have restrained our discussion.

Interestingly, our results are very similar to our original *eLife* paper that this builds on as a Research Advance, which used a very similar breathing paradigm and noise correction techniques, and also did not produce any dorsal PAG activity. It is very possible that the specific breathlessness paradigm we employed, and the enforced constraint on the available behavioural responses whilst subjects were in the MRI scanner may have influenced these results and limited dorsal PAG activity. This situation differs vastly from many studies in freely-moving animals with implanted electrodes, or direct-stimulation studies of PAG columns in decerebrate animals.

Although the aforementioned Satpute paper does mention dmPAG activity in the text of its results, the main figure and data presented in that paper does not support this; instead demonstrating main activity in the vlPAG and bordering lPAG whilst subjects viewed threatening images, with no evidence for dmPAG activity. Additionally, the PAG subdivisions employed in the Satpute paper are also based upon a cylindrical nucleus that rely on a ‘best guess’ as to how the structure might be subdivided, rather than on the literature, as there was no literature detailing the substructure of the human PAG at the time this paper was published. We know that the PAG has separate, well-differentiated (non-PAG) nuclei on the ventromedial border that are included in the Satpute et al., PAG, and their PAG nucleus also does not identify a dlPAG column. We have now included a section within the limitations section of the Discussion to address the dorsal PAG (or lack thereof) in our study, with the following text:

“This study has used 7 Tesla fMRI to investigate differential activity and functional connectivity of the human PAG columns in association with perceived threat, and in this instance the respiratory threat of breathlessness. […] Alternatively, these findings may represent interspecies differences in PAG activity and/or connectivity when responding to the threat of breathlessness.”

*15) Main text, second paragraph: What are the premotor areas in the ponto-medullary reticular formation? How is this circuitry built? See the comments regarding connectivity above.*

*Maybe it might be helpful for the reader to add a summary diagram, showing the incoming and out-going pathways of the different parts of the PAG. The present version of this paper generates a lot of question marks.*

Our mention of the ponto-medullary reticular formation was to simply highlight potential comparisons that could be made between recent work in animals (Tovote et al., 2016) and human work. However, we appreciate that this was by no means the primary focus of our work, and our scanning sequence was not optimised to perform this. Furthermore, the analysis that demonstrated these findings used unthresholded statistics, therefore this lower brainstem analysis would require considerable caution in its interpretation.

The regions identified by Tovote and colleagues (2016) as a distinct premotor region in the medulla was specific to the magnocellular nucleus, although current fMRI methods do not have the spatial resolution or contrast required to isolate specific medullary nuclei.

We have therefore decided that it would be more appropriate not to include these results within this manuscript, as these were primarily exploratory data based on unthresholded results. We would not wish to be misleading to any reader less familiar with the stringency needed for statistical analysis in fMRI. However, we are happy to include these results only in the supplementary material if this would be preferable to the reviewers. We feel further expansion of these findings would be better served by a more definitive study focused upon brainstem connectivity.

*16) There are species-specific differences in the functional topography, which the authors need to consider in interpreting the results from humans, using imaging. Topographic chemical or optical stimulation have led to detailed insights in columnar specificity that is not entirely consistent with this human study. Both Bandler, & Carrive labs (Carrive, 1993, an important citation missing)/Subramanian and Holstege labs have shown the lateral PAG to be involved in tachypnea. In the cat this leads to flight and as per Bandler/Carrive lab findings, also fight. While in the rat, the lPAG induced tachypnea may well correspond to dyspnea or fright. Freezing however is more caudal and ventrolateral (Subramanian & Holstege 2013a/b, 2014) characterised by distinct apnea/ breath-hold. One could assume that the increase in anxiety levels could affect fright towards freezing per se, thus a breath-hold or apnea, which would then increase the activity in the vlPAG rather than lPAG. However, the results reported by the authors are contrary. Could the authors comment on this? In particular, in light of the experimental approach that provides only limited insights into the causal relationship to respiration as raised in one of the first comments.*

Thank you for this interesting discussion point. Studying humans gives us the unique opportunity to ask individuals their subjective ratings of emotions such as anxiety, rather than inferring these from physiological and behavioural measures in animals. Additionally, as well as potential species-specific differences in functional topography, a further difference between testing animals and humans is that we are not directly stimulating the PAG columns and measuring their response, but rather inducing a behavioural change and measuring associated signal in the individual PAG columns.

Despite these differences, the cross-species tachypnea referred to in the examples provided by the reviewers is an interesting and relevant parallel to the lPAG-linked active effortful breathing required to overcome the inspiratory resistance in this study, whereas the ‘freezing’ response in the vlPAG parallels the vlPAG activity we observed during anticipation. As noted, we observed heightened anxiety for breathlessness cues than non-breathlessness cues. However, as both the anticipation and breathlessness were (unavoidably) indicated by the same conditioned symbol, there was no way of separating anxiety associated with anticipation with that for breathlessness.

Therefore, we are unsure whether the anxiety recorded upon presentation of the shape immediately following cessation of the experiment was isolated to the anticipation or breathlessness periods, or both. While it would be tempting to use reverse inference to infer that the vlPAG activity during anticipation may indicate a stronger association with anxiety than during breathlessness itself (which was associated with lPAG activity), we are unable to tease this apart using the current study design.

Much work is still to be done in relating animal and human PAG investigations, and working together across species to understanding how these findings translate and/or differ. We have now raised this point in the limitations section of the Discussion:

“Whilst the use of ultra-high field scanners affords us higher-resolutions that facilitate exploration of the subdivisions of the human PAG, there are some constraints within fMRI scanning and analysis that need to be addressed. […] Therefore, an area of interest (such as the individual columns of the PAG) cannot be viewed in isolation when using fMRI, nor its activity identified as causal to the outcome of the task.”

*17) Anticipation & breathlessness: Was any activity seen in the bed nucleus? Particularly when the central nucleus of the amygdala together with the lateral bed nucleus of the stria terminalis projecting strongly to the PAG, one would assume a strong activation of the bed strialis. Was this seen?*

There was no significant activity in the bed nucleus of the stria terminalis in either anticipation or breathlessness.

*18) The authors report on the resting functional connectivity of the vlPAG to the ACC. Did the authors see any PAG columnar interactions with the ACC during either breathlessness or its anticipation?*

The functional connectivity observed between the vlPAG and ACC was limited to rest, and was not seen with either of the PAG columns during task-based functional connectivity. We are aware there will be many other functional communication pathways that were not identified in this analysis, these may not be as tightly time-locked to columnar PAG activity and thus do not appear using these types of functional connectivity methods. We have now mentioned this in our limitations section:

“This work attempts to quantify ‘functional connectivity’ between the active PAG columns and cortex, both at rest and during breathlessness anticipation and perception. […] Therefore, we are currently unsure of the driving, causal centres within these functionally-associated areas, and may be insensitive to functional associations that are too temporally asynchronous or dynamic for our statistical thresholds to identify.”

*19) Subsection “lPAG in the fight/flight response to threat” seems a bit far-fetched to me, particularly the inference of 'plausible pathways' for goal-directed behaviours. Does an 'escape' response either via flight or fright involve 'goal directed behaviour"? It would seem more an innate response to me, although the functional connectivity with the caudate is of significant interest no doubt.*

Thank you for this comment – we have now adjusted this statement as we agree that we cannot disentangle goal-directed from innate pathways from this study design, and it is very possible this is an innate response. We have now altered the text as follows:

“This profile demonstrates plausible pathways for involvement in active responses through functional sensorimotor connections to the primary motor and sensory cortices and caudate nucleus […]”

*20) Holstege & Hopkins showed in the cat that wiring includes amygdala-bed nucleus triangulation into the PAG. It's a bit of a long shot to suggest that anxiety-encoding involves just the amygdala-PAG axis in humans (that to cross-referenced to work on transgenic mice) despite the pronouncement of the terminalis area.*

Thank you for this comment – we were not in any way trying to infer that this was the only pathway involved in anxiety-encoding, and agree that it is very likely other areas such as the bed nucleus are likely playing a vitally important role. As mentioned, due to functional connectivity measures relying heavily on time-locked signal fluctuations, it is possible that these types of connections are overlooked in these analyses. We hope that by addressing this issue in the limitations section this may now be clearer for readers. We have also added a reference to the Holstege and Hopkins paper and mention of the bed nucleus into the Discussion as a point of note for further research in humans. This text is now as follows:

“These results reveal a potential anxiety-encoding pathway through the amygdala during the perception of the breathlessness stimulus (Davis 1992, Stein 2007). This pathway may incorporate into both existing neurocognitive anxiety models of prefrontal-amygdala top-down contributions to threat responses (Bishop 2008, Bishop 2007), and amygdala – bed nucleus of the stria terminalis triangulation with the PAG previously shown in animals (Hopkins 1978, Gray 1992).”

*21) Anticipatory respiratory threats would involve learning mechanisms which brings in the hippocampal-amygdaloid interaction. Was any lighting up seen in any of the hippocampal regions?*

Indeed, we did see activity in the hippocampus and amygdala during both anticipation and breathlessness (shown in Figure 1). This was also observed in our previous *eLife* paper (Faull 2016), and we observed similar (smaller) findings in the hippocampus in a separate study during anticipation of resistive loading with saline and opioids (Hayen, NeuroImage – under review). However, no significant functional connectivity between PAG columns and the hippocampus were observed during anticipation or breathlessness.

*22) While the Tovote et al. optogenetic study provides salient details of the amygdala-PAG interaction, their identification of a group of cells what they call "magnocellular nucleus of the medulla" seems strange. The area they identify in the medulla points more or less to ventrolateral medullary cells, a region involved in cardiorespiratory control, rather than locomotor control.*

We agree that the medullary projection areas from the PAG in the Tovote et al. study are somewhat ambiguous, despite their identification of the magnocellular nucleus. As this was not the primary aim of this study and simply an exploratory observation, we have now removed these findings and their discussion as previously explained in point 15 above.

*23) Studies by the Subramanian lab demonstrated direct modulation of the respiratory cells by the PAG in inducing various respiratory effects. Schimitel and Gargaglioni labs also suggested hypoxia-induced reduction and avoidance underplays freezing behavior. They also involved the VLM cardiorespiratory cells. In another important study, Oka et al. (2008) demonstrated that GABA-ergic neurons in the central nucleus of the amygdala possibly influence the PAG-NRA pathway in mediating fear. Thus, the authors need to carefully pitch their reasoning for examining the ponto-medullary structures.*

As with our previous comments, we agree that including the PAG-medullary interactions needs careful reasoning and wasn’t the primary purpose of this study. It has now been removed from the manuscript as per point 15 above.

*24) Subsection “Intra-PAG connectivity and interaction between fight/flight and freeze responses”, combines species across spectrum from the rat (Jansen et al.) to the transgenic mice (Tovote). The columnar structure of the PAG as defined in cat and to a certain extent in rat has been simply incorporated to the transgenic mice. The rostrocaudal extension of the PAG defines at large the columnar structure (Carrive) identified mainly via behavioural and autonomic topography. In the mice, this fundamental information is forthcoming. The discussion on this section is largely based upon reciprocal inhibition suggested based on anatomical wiring (Jansen) and the study by Tovote. Lack of concrete human data, means this can be simplified rather than relying heavily on cross-species animal work.*

Thank you for this helpful suggestion. We have now considerably simplified this section, and highlighted the difference between what is known from animal models and the lack of human data to date. The text now reads as follows:

“While the vlPAG and lPAG appear to have very different functions within the threat response to breathlessness, it is possible that inter-columnar communications may allow appropriate stimulus encoding and subsequent behavioural responses. […] Therefore, the apparent reduced connectivity to motor structures during ‘freeze’ may be via inhibition of the lPAG and its downstream connections, inhibiting the ‘fight/flight’ to instigate freezing.”

*25-28) The reviewers also had various methodological comments that need to be addressed:*

*25) The authors make many specific methodologic choices in their pre-processing. Some of these are less common (for example, a 2mm Gaussian kernel) and some have their believers and non-believers (for example, field unwarping). The reviewers appreciate the reference to a previous recent paper by the same first author, but the authors should also comment (briefly) on some of the more prominent choices. As well, if there are any specific choices made on the basis of imaging at 7T strength it would be helpful to contextualize this.*

Thank you for this comment. We have now been through the Methods section and elaborated on the choices within our analysis. These are as follows:

We used 2mm spatial smoothing (as opposed to 6 mm or greater that is more common) as we wanted to better define small structures, such as the columns of the PAG. The increased signal afforded by 7 Tesla imaging allowed us to improve spatial resolution from the typical 3x3x3xmm+ to 2x2x2mm, and thus we wanted to minimize smoothing. Furthermore, smoothing can even ‘smooth out’ activity isolated to small structures within the brain. However, this approach is conservative as (generally speaking) larger smoothing improves statistical power.

“The following preprocessing methods were used prior to statistical analysis: motion correction and motion parameter recording (using MCFLIRT: Motion Correction using FMRIB's Linear Image Registration Tool) (Jenkinson et al., 2002)), removal of the non-brain structures (skull and surrounding tissue) (using BET: Brain Extraction Tool (Smith, 2002)), spatial smoothing using a full-width half-maximum Gaussian kernel of 2 mm, and high-pass temporal filtering (Gaussian-weighted least-squares straight line fitting; 120 s). Spatial smoothing was limited to 2 mm, to maintain the improved functional resolution (2x2x2 mm) afforded by 7 Tesla imaging. Minimal spatial smoothing, whilst conservative, reduces blurring between small structures, such as the columns of the PAG.”

We have now added references to the distortion-correction explanation to provide the evidence on which we implemented the use of this tool. Distortion-correction is the accepted standard practice in neuroimaging using FSL, as this allows improves registration to structural scans and onwards to the group template or standard brain, especially in the brainstem where registration is difficult due to large distortions.

“Distortion correction of EPI scans was conducted using a combination of FUGUE (FMRIB's Utility for Geometrically Unwarping EPIs) (Holland et al., 2010; Jenkinson, 2001; Jezzard, 2012) and BBR (Boundary Based Registration; part of the FMRI Expert Analysis Tool, FEAT, version 6.0 (Greve and Fischl, 2009)). Distortion correction is particularly important in areas of the brain that suffer considerable distortions due nearby bony interfaces and proximity to sinuses, such as occurs in the brainstem.”

We have also explained our reasoning for using the combined noise-correction techniques of ICA-denoising and RETROICOR. The main sources of physiological noise in fMRI relate to the respiratory and cardiac cycle (Brooks et al., 2013; Glover et al., 2000; Harvey et al., 2008). Whilst cleaning imaging data of unwanted noise, the main caveat for noise correction in general is that any neural signals that are aliased to the cardiac and respiratory cycles will be lost. RETROICOR uses continuous measures of cardiac and respiratory cycles during a scan to regresses out these known quantities. However, it requires hardware investment and technical expertise at the time of scanning. ICA decomposition requires training data that has to be classified manually, which is time consuming and can be subject to experimenter bias. The benefit of using a combination of RETROICOR and ICA decomposition is a comprehensive noise correction, but this has the caveat of requiring an even more complex analysis to ensure that noise is not reintroduced.

“Due to the location of the PAG and its susceptibility to physiological noise (Brooks et al., 2013), rigorous data denoising was conducted using a combination of independent components analysis (ICA) and retrospective image correction (RETROICOR) (Brooks et al., 2013; Harvey et al., 2008), as previously described (Faull et al., 2016b). […] Physiological recordings of heart rate and respiration from respiratory bellows were transformed into regressors (3 cardiac, 4 respiratory harmonics, an interaction term and a measure of respiratory volume per unit of time (RVT) (Harvey et al., 2008) corresponding to each acquisition slice, and the signal associated with this noise was isolated using a linear regression, adjusted for any interaction with previously-identified ICA noise components and then subtracted from the data.”

We have included an explanation for our haemodynamic response modelling, which uses an optimal basis set rather than a standard gamma waveform. The haemodynamic response function (HRF) varies between brain regions and individuals (Handwerker et al., 2004), and the differences in HRF between small structures such as the PAG and large areas of cortex are not yet known. Therefore, instead of using the fixed standard gamma waveform, we used an optimal basis set of three waveforms (FLOBS: FMRIB's Linear Optimal Basis Sets, default FLOBS supplied in FSL (Woolrich et al., 2004a)). This allows us to have a more flexible HRF model that will account for differences in the HRF between regions of the brain within a single model. Although this approach may more accurately account for varying HRFs, it may lead to a small underestimation of the effect size as it does not account for changes in amplitude that may be seen when an HRF is temporally shifted. Therefore, it is a conservative estimation, as it will not over-estimate the HRF and thus not introduce erroneous false positives.

“Functional voxelwise analysis incorporated HRF modeling using three optimal basis functions (FLOBS: FMRIB’s Linear Optimal Basis Set), instead of the more common gamma waveform. […] This HRF flexibility is especially important when modelling both small regions (such as the PAG) and large areas of cortex in a single model, where the HRF likely differs across these regions.”

*26) The authors here perform essentially a cluster based task-fMRI analysis. Recently, these methods have come under substantial scrutiny in the wider neuroscience community due to their vulnerability to serial autocorrelation. The reviewers suggest that the authors acknowledge this, and explain in detail how their use of correction in FILM, as well as the significance thresholds they employ in their statistical analysis, adequately address these concerns. For example, the reviewers noted that the authors use a significance threshold of p<0.05 to perform cluster testing.*

*Please define correction methods used during significance testing.*

We appreciate that this is a hot topic within the field that needs to be addressed. While parametric analysis was used for the whole-brain exploration, we have now elaborated on the correction techniques used and explained how both FILM local autocorrelation correction and minimal spatial smoothing helps to control inflations in false positive rate in FSL.

Furthermore, we have elaborated that any small-volume analysis of the PAG was conducted using permutation-testing (consistent with our previous *eLife* paper – Faull 2016), which has been shown to robustly control for false positives. The text in the methods is now as follows:

“For whole-brain results, Z statistic images were thresholded using clusters determined by Z > 2.3 and a (family-wise error (FWE) corrected) cluster significance threshold of p < 0.05. Recent scrutiny of fMRI statistical methods has raised concerns over cluster-based thresholding when combined with parametric testing (Eklund 2016). […] Permutation testing has been shown to robustly control for appropriate false positive rates with both voxelwise and cluster thresholding (Eklund 2016).”

*27) There are many FSL-specific abbreviations used in the Methods section. They have the cumulative effect of diminishing the overall clarity and understanding. The reviewers suggest defining that these are toolboxes within FSL and specifying what they are before acronyms are used. As well, a little too much is assumed in terms of how various toolboxes in FSL perform these various components of processing and analysis. In our community, various choices of software exist, and each involves some embedded methodologic choices. It is helpful to at least briefly but clearly state what is implied by the choice of each major step in the FSL pipeline.*

Thank you for this helpful comment. We have now made sure that each abbreviation is explained in full so that the function and implication of each step in the analysis pipeline is apparent and clear. We have also added further details to the Supplementary material (which has been mentioned in the Methods section), with links provided to FSL documentation online.

“Data preprocessing

The ‘cleaned data’ described in Supplementary Table 3 is the data following preprocessing and noise correction steps, prior to being fed into the statistical analysis for task or resting state. […] Powers et al. (2012) use a threshold of 0.5% BOLD signal change as an acceptable value for DVARS measurements, and our data shows each subject’s motion as below this threshold following rigorous preprocessing.”

*28) The reviewers encourage the authors to provide a spreadsheet as a supplement including, on a per subject basis, the regressors used as well as measures of head motion, with brief summary statistics (e.g. mean and SD).*

We have now included Figure 1—figure supplement 1 that demonstrates the general linear model used in the task analysis and task-based functional connectivity analyses at the single subject level. The motion regressors were incorporated into the ICA-denoising step in the analysis, which we have now outlined in both the methods and in the legend of this figure supplement.

We have also now included a table ([Supplementary-material SD2-data]) that provides measures (mean and standard deviation) of head motion per subject, both before and after all preprocessing steps to clean the data (using a DVARS metric of% change in BOLD). The Powers et al. (NeuroImage, 2012) referenced a threshold of 0.5% change in BOLD for outlier detection, and all subjects’ calculated mean values following preprocessing were below this threshold.

[Editors' note: further revisions were requested prior to acceptance, as described below.]

*Essential revisions:*

*1) One set of comments relates to the handling of motion correction. The authors have regressed out variance they identified with motion. It will be important to specify whether this was done at the first or second level analysis.*

Thank you for highlighting this point. We have now adjusted our Methods section to clarify at what level of analysis we have regressed out residual motion. The text now reads as follows:

“Head motion regressors calculated from the motion correction preprocessing step (using MCFLIRT in this case) were regressed out of the data alongside the ‘noise’ components identified during ICA denoising, during preprocessing of the functional scans prior to first level analysis.”

*However, the bigger issue is whether this study can exclude residual motion after denoising? This can be accomplished, for example, by including motion regressors in the statistical analysis at the second level. Since the authors have already calculated their DVARs measures this should be relatively fast. They would then have the opportunity to declare their significant results were unaffected by residual motion.*

The reviewers have raised an important point, as we agree that fMRI (and especially functional connectivity) is susceptible to motion-induced artefacts. At this additional request to incorporate the calculated residual motion as a regressor in the highest level of the analysis, we have re-run all of our analyses with this change (functional activity as well as functional connectivity). All results remained remarkably similar to those previously reported, with small changes in the functional connectivity of the lPAG both during breathlessness and at rest. The addition of an extra explanatory variable in the model results in a loss of power due to changes in degrees of freedom; however, as the issue of motion is so topical in functional connectivity, we have adopted these more stringent results. We have now changed the Methods section (see text below) to include a description of this regressor, and we have replaced all figures and any necessary result descriptions to accurately represent these results. We hope that these rigorous motion correction measures will increase the validity of our results in the eyes of the reader, and we thank the reviewers for their suggestions.

“An additional residual motion regressor was included in this group analysis, consisting of subject-specific DVARS values (see Supplementary material), to ensure the results were independent of any effects of residual motion.”

*2) In response to comment 14, the authors state: "The Satpute paper does mention dmPAG activity in the text of its results, the main figure and data presented in that paper does not support this; instead demonstrating main activity in the vlPAG and bordering lPAG whilst subjects viewed threatening images, with no evidence for dmPAG activity. Additionally, the PAG subdivisions employed in the Satpute paper are also based upon a cylindrical nucleus that rely on a 'best guess' as to how the structure might be subdivided, rather than on the literature, as there was no literature detailing the substructure of the human PAG. We know that the PAG has separate, well-differentiated (non-PAG) nuclei on the ventromedial border that are included in the Satpute et al., PAG, and their PAG nucleus also does not identify a dlPAG column. We have now included a section within the limitations section of the discussion to address the dorsal PAG (or lack thereof) in our study, with the following text".*

*However, the text (in subsection “Limitations”, first two paragraphs) added to the manuscript does not really address the issue of dorsal PAG (or lack thereof)! The authors have merely stated a well-known limitation of the fMRI method. Indeed, fMRI method does not provide topographic specificity of the columns, or the effect seen can be interpreted as causal to the stimuli. Stating this does not answer the question in the context of the dorsal PAG.*

*The authors should discuss in the Discussion section the reasoning in terms of the structure of the human PAG (cylindrical vs. any other published /assumed), the lack of border differentiation of the human PAG, and why this was seen, noted or inferred on in other studies (say Satpute et al) and not in the present study. The current results are similar to the earlier eLife paper and would just remain an incremental research advance if avenues such as the lack of activation of the dorsal PAG, a major center implicated in subsequent sensory processing during and following a somatic/autonomic outflow is not discussed adequately.*

Thank you for this comment. We have now added in a separate Discussion paragraph to directly acknowledge the lack of dorsal PAG activity in our current experimental paradigm. The text reads as follows:

“Interestingly, in both this study and the original investigation (Faull et al. 2016) we did not identify dorsal PAG activity during any of our tasks. […] Therefore, whilst we were unable to identify dorsal PAG activity within our specific experimental task, further work is required to more fully elucidate the roles of the human PAG subdivisions across a range of threatening stimuli.”

*3) Related to comment 16: We agree that "studying humans gives us the unique opportunity to ask individuals their subjective ratings of emotions such as anxiety, rather than inferring these from physiological and behavioural measures in animals". Irrespective of whether one examines by direct stimulation of PAG columns or assessing signal activation of the columns indirectly, the authors are dealing with a set of highly specialized circuitry, topographically organized for appropriating specific physiological and behavioural responses. Given the non-specificity of the fMRI method vs. highly specific animal methods, I would consider it more important to discuss the findings in humans against animal observations (with appropriate citations, as key references are missing) for inference on PAG function regarding respiratory threat in humans. We suggest that the authors add a separate paragraph to examine human findings against animal studies that have examined the physiology and behavioural responses to respiratory threat.*

Thank you for this suggestion. We have now included a Discussion section to directly compare the respiratory-related animal findings from PAG stimulation to our work in humans with a respiratory threat. The text now reads as follows:

“Respiratory evidence from animal models

For the specific respiratory response to a breathlessness threat, we can draw comparisons between the findings from this study and previous respiratory investigations in animals. […] Therefore, it is clear that further investigation is needed to understand the intricate control of respiration by the PAG in the face of a both conditioned respiratory threat such as breathlessness, and towards other conditioned threats.”

*4) Related to the above and comment 23. The various (animal) studies that we listed deal with breathlessness that can be induced from the PAG, its connotations to suffocation alarm, fear and perception of respiratory threat. Since the current study deals with the threat of breathlessness (and not a generic fear mediation circuitry as investigated by Tovote et al), a discussion linking the human findings to animal studies in the context of respiratory threat is imperative. Particularly when the authors infer upon the inhibition of lPAG and its downstream connections inhibiting the flight/fight to instigate freezing which are clearly elucidated in animal models.*

Thank you for this suggestion. As per the comment above, we have now included a discussion that directly addresses the animal vs. human literature. We have also included further animal citations in our discussion of the intra-PAG connectivity, which now reads as follows:

“While the vlPAG and lPAG appear to have very different functions within the threat response to breathlessness, it is possible that inter-columnar communications may allow appropriate stimulus encoding and subsequent behavioural responses. […] Therefore, the apparent reduced connectivity to motor structures during ‘freeze’ may be via inhibition of the lPAG and its downstream connections, inhibiting the ‘fight/flight’ to instigate freezing.”